# Characterizing the dynamic movement of thunderstorms using VLF/LF total lightning data over the Pearl River Delta region

Si Cheng[1], Jianguo Wang[1], Li Cai[1], Mi Zhou[1], Rui Su[1], Yijun Huang[1], Quanxin Li[1]

[1] Electrical Engineering and Automation, Wuhan University, Wuhan, 430072, China

*Correspondence to*: Jianguo Wang (wjg@whu.edu.cn); Li Cai (caili@whu.edu.cn)

**Abstract.** This paper reveals the dynamic movement characteristics of thunderstorms using total lightning data obtained from the VLF/LF location measurement. Eight thunderstorms, which were evenly distributed in the morning, midday, afternoon and evening, are selected to compare the different kinematic features over the Pearl River Delta (PRD) region in the south of China

from 17 May to 23 May 2014. The connected-neighbourhood labeling method is used to identify lightning clusters and obtain the centroids. Significant characterization parameters are put forward as metrics to reveal the kinematic features of thunderstorms, including the duration time, valid area (VA), movement velocity, movement direction, and farthest distance in longitude and latitude during the life cycle of the storm. A common trend is that the storms initiate in the west of the PRD region, moving to the east and disappearing after the thunderstorm travels around 106.5 km in longitude. There are two kinds

of distributions to depict the property of the valid area, which are one-peak distribution with the maximum in the mature stage and two-peak distribution with a relatively smaller peak in the early time of the storm. The velocity does not show the same trend as the variation of VA which shows a steady increase or decrease during the lifetime of thunderstorms. The biggest VA and highest velocity are 891 km$^2$ occurred on the evening of 17 May and 204.8 km h$^{-1}$ occurred on the morning of 20 May. The 19 May evening storm was the weakest, with the maximum VA and velocity being 253 km$^2$ and 115.3 km h$^{-1}$, respectively.

The motion of eight storms shows a distinct pattern, as the spread of direction distributes tightly in the range of 0°- 180°. The movement characteristics of thunderstorms and the associated parameters may help to improve the nowcasting and forecasting system of thunderstorms in the warm season.

## 1 Introduction

As one of the noticeable weather events in nature, the thunderstorm and its dynamic movement are of great interest in

engineering applications and in the analysis of interactions between lightning and Earth's atmosphere (Kandalgaonkar, 2005; Zeng et al., 2016; Jayawardena and Mäkelä, 2021). Lightning has been the subject of intense scientific research because of its close relationship with severe weather and convective rainfall, thus posing a great threat to lightning-sensitive facilities, such as airports, energy sector infrastructure, maritime assets and military bases (Fankhauser, 1971; Lee and Passner, 1993;

Cummins et al., 1998; Keenan et al., 2000; Villarini and Smith, 2013), and the environment (Krider et al., 1980; Lee, 2017).

Thunderstorm identification, tracking and short-range forecasting for the future clusters through the lifetime would constitute the basis of severe weather warning operations.

A high-resolution observation system and automatic algorithms applied to these data are needed to detect, evaluate, track, forecast thunderstorms and obtain quantitative information including the position, size, path and velocity of the cell (Bonelli and Marcacci, 2008). The evaluation of thunderstorm occurrence and characteristics is conducted through lightning data (Kohn

et al., 2011; Meyer et al., 2013b), radar data (Del Moral et al., 2018; Miller and Mote, 2017), or the combination of both (Bonelli and Marcacci, 2008; Lu et al., 2021; Meyer et al., 2013a; Rigo et al., 2010). The lightning detection methods include ground-based systems, such as the LINET (Betz et al., 2008), Earth Networks Total Lightning Network (ENTLN, (Ringhausen and Bitzer, 2021), ZEUS (Kohn et al., 2011) and Lightning Mapping Array (LMA,(Weiss et al., 2012)), and the space-based satellites, such as the Lightning Image Sensor (LIS,(Chronis and Koshak, 2017; Zhang et al., 2019)), Optical Transient

Detector (OTD,(Buechler et al., 2000; Christian, 2003) and Geostationary Lightning Mapper (GLM,(Rutledge et al., 2020). The reflectivity images provided by the Weather Surveillance Radar (WSR) are commonly used to identify those areas of convection in a certain time interval. With the ability to capture the spatial and temporal development of thunderstorms, radar data with high resolution can provide a detailed analysis of the variation and movement of convection storms (Muñoz et al., 2018). After the recognition of high reflectivity areas or high lightning density areas with automatic algorithms, the motion of

thunderstorms can be tracked and extrapolated.

Statistics and comparisons of storm attributes, such as the direction and speed of movement or cell sizes and severity, are studied by researchers. Different synoptic conditions are typically in correspondence with specific cell characteristics based on the characteristics of convective cells in central Europe (Wapler and James, 2015). The connections between various cell attributes and synoptic patterns are significant. For example, storms associated with the broadly westerly

flow are more likely to have high cell speeds and a relatively narrow distribution of cell directions. The large-scale weather conditions with lower average cell speeds have a higher tendency to produce hail. The comparison of storm speed done as a function of the month of the year is conducted between the winter and summer storms (Kohn et al., 2011), showing that the summer ones are much faster than the winter ones, which might be in contradiction with the assumption of faster storms in the winter due to the strengthening of the jet stream at this time. The possible reason is that the summer

storms are convective in nature and therefore stronger and faster. The lightning parameter's relationship to Hurricane Harvey's intensification is conducted based on a merged lightning data set consisting of lightning detected by the ground-based ENTLN and space-borne GLM (Ringhausen and Bitzer, 2021). There was a large increase just prior to rapid intensification in the rainband and eyewall region, with the flash duration, the number of events, and groups comprising a flash showing the largest increases. However, up to now, there have been few formal studies that individually analyze

such fundamental kinematic characteristics of every single thunderstorm. It is the objective of this study to address the aforementioned.

The purpose of this paper is to provide a comprehensive kinematic feature of eight thunderstorms over the PRD region from the perspective of the distribution of total lightning data obtained from the VLF/LF Foshan Total Lightning Location System. We conduct a detailed analysis of the temporal and spatial evolution of eight cases that occurred within a week from 17 May to 23 May 2014. The 8-adjacent connected-neighbourhood labeling algorithm is applied for the identification of clusters during the lifetime of thunderstorms, through which the centroids and valid areas are figured out every 12-minute time interval. To characterize the spatial evolution, five parameters are put forward to quantify the movement of clusters in various periods of a day. The result shows that there is a clear pattern in terms of the transition of thunderstorms in this region.

## 2 Instrument and methodology

### 2.1 Foshan total lightning location system

In the summer of 2013, a 3-dimension Foshan total lightning location system (FTLLS) was installed in Guangdong Province, China (Cai, 2013; Cai et al., 2019). Based on the electromagnetic environment, surrounding buildings, terrain conditions, communication conditions, etc., nine stations were developed to detect lightning events over the PRD region, as shown in Figure 1. The distance between each station is generally 10 to 40 km and the detection range is more than 100 km. The coverage of the entire station network is about 1000 km$^2$, which can effectively cover the lightning activity area in PRD. The DTZ, MCZ and JAZ stations are far apart, forming a longer baseline, which can effectively improve the locating accuracy, while the remaining sub-stations are densely distributed.

All the nine stations are installed on the roof of the building of subsidiaries of Foshan Electric Power Company, China Southern Power Grid. The power supply is achieved via the 220 VAC power network. Wide-band electric field measuring systems with a 3 dB bandwidth from 200 Hz to 500 kHz are employed to measure the lightning electromagnetic impulses. A three-dimension location algorithm is utilized in FTLLS. The location information contains the height of lightning, which can be applied to identify the discharge types. The characteristic parameters of the radiated electric field waveform produced by different types of discharge events can also be served as a discrimination method. Each current pulse detected by the FTLLS is defined as a lightning event. The classification of cloud-to-ground events (CG), intra-cloud events (IC) and narrow bipolar events (NBEs) can be accomplished by a combined analysis of those parameters from FTLLS (Cai et al., 2019). The system can provide real-time lightning data to the electric utility industry, which mainly includes the time of lightning strokes, three-dimension location, peak current, rise time, fall time, pulse width and signal-to-noise ratio. Based on the Monte Carlo simulation, the two-dimensional horizontal location error is basically less than 100 m, and the vertical error (altitude) is less than 200 m when the lightning event occurs within the network. Li et al. (2021) examined the detection efficiencies using the directly measured current data at the triggering lightning site. The result shows that the detection efficiencies of FTLLS for flashes and return strokes were 87.5% and 93.0%, respectively. The validation of the system has been guaranteed through the

comparison of rocket-triggered lightning experiments and the application of transmission lines (Cai et al., 2019; Wang et al., 2019).

## 2.2 Data and Methodology

During the monsoon period (in May on average) in the PRD region, the South China Sea Summer Monsoon (SCSSM) enhances the precipitation owing to the southwesterly monsoon flow, especially the southwesterly low-level jets, carrying abundant water vapor to South China (Bei et al., 2002; Chen and Luo, 2018). More than 50% of heavy rainfall events in South China occur in April to June period, in which precipitation is primarily related to fronts and monsoon flows (Wu et al., 2011). Persistent heavy rainfall occurred from 17 May 2014 to 23 May 2014, especially in the central and eastern parts of the PRD

region. It was reported that 62 automatic weather stations recorded heavy precipitation of more than 100 mm on 17 May, while Huizhou and Shenzhen stations recorded 24 h rainfall of 377.9 mm and 274.6 mm (Bingzhi Zheng, 2015). Severe thunderstorms occurred in the PRD region with a record-breaking 24 h rainfall of 477.4 mm starting from 2000 LST (local standard time = UTC + 8 h) on 22 May during this heavy rainfall week. The hourly precipitation in the Conghua station surpassed 60 mm at 1300 LST on 23 May (Zhongqing Liang, 2015; Xinyu Zhou, 2017). Figure 2 showed general radar

characteristics and lightning distributions of thunderstorms from 11:36 to 14:36 and 17:48 to 20:36 on 17 May. The radar data is obtained from China's Weather Surveillance Doppler-1998 (WSR-98D) at Guangzhou. The lightning events are located in the area with radar reflectivity higher than 30 dBz, which has been verified as the threshold in the south of China (Xu et al., 2010). Consecutive precipitations within one week include extremely severe thunderstorms and relatively mild thunderstorms which serve as great cases for comparison.

All geographical plots in this paper are created by counting lightning events within $0.01° \times 0.01°$ grid boxes, corresponding to an approximate resolution of 1 km over the Pearl River Delta region (112°-115°E, 22°-25°N). If a larger box is employed to count lightning events, more lightning events will be contained within each box, leading to ambiguous cluster recognitions. Conversely, smaller resolution results in plenty of empty boxes, mistakenly separating the thunderstorm clusters. The time interval of 12 min is twice of the Doppler Radar scans, with which the routes of thunderstorms can be tracked precisely without

losing kinematic features.

Connected-neighbourhood labeling is applied to grid boxes to identify lightning clusters. Connectivity means that a connected path can be formed between two boxes in the area. From the perspective of digital images, connectivity can be classified into two types: (1) 4-adjacent connected-neighbourhood labeling, which refers to starting at any pixel position in a collection or area and searching from four directions (left, above, right, below) of the pixel, any other pixels can be found in

the collection or area; and (2) 8-adjacent connected-neighbourhood labeling, which is the same as 4-adjacent, but adds four diagonal positions (Miller and Mote, 2017; Xue et al., 2019; Zan et al., 2019). Connected neighbourhood labeling is to give each connected area a unique number during the search process. In this paper, the second type is adopted to automatically identify lightning clusters.

The process of thunderstorm visualization is shown in Figure 3. The analysis area is divided into 0.01°×0.01° grid boxes, corresponding to the geographic area of 1 km$^2$ approximately. The number of individual lightning events is counted within each grid box at 12-minute intervals, on which we rely to draw the lightning density map. Setting one lightning event as the density threshold, the box with more than an event can be defined as a valid box. Using the 8-adjacent connection neighbourhood labeling algorithm, we can figure out the number of valid boxes in each thunderstorm cluster. As the area of each box is 1 km$^2$, we define the number of valid boxes as the valid area (VA). To better capture the main spatial movement of the thunderstorm, clusters less than 25 km$^2$ are removed based on the scale of thunderstorms in the PRD region. The less strong thunderstorms are not considered in the article since it is the strong thunderstorms that pose great damage. The final area of each cluster is substituted by an equivalent circle (EC). Equation (1) is the conversion formula of VA and the radium of equivalent circle (REC), which is used to draw the ground motion map in the following section.

$$\text{REC} = \sqrt{\frac{\text{VA}}{\pi}} \tag{1}$$

Taking the proportion of lightning frequencies in each grid to the total number of lightning in all effective grids as the weight, the longitude and latitude coordinates of discharge centroid (C) are obtained by the weighted average of each grid within the valid area. The expressions are shown in Eq. (2) and Eq. (3).

$$C_{\text{lon}} = \sum_{N=1}^{VA}\left(\frac{\text{the number of events in Grid N}}{\text{total events in valid area}} * N_{\text{lon}}\right) \tag{2}$$

$$C_{\text{lat}} = \sum_{N=1}^{VA}\left(\frac{\text{the number of events in Grid N}}{\text{total events in valid area}} * N_{\text{lat}}\right) \tag{3}$$

where Clon is the longitude of discharge centroid in the density diagram, Clat is the latitude of discharge centroid, Nlon is the longitude of one gird in all the effective grids in the density diagram, Nlat is the latitude of one gird in all the effective grids.

As the coordinates of discharge centroids within a time interval are obtained, two clusters whose discharge centroid is less than 10 km merge as one cluster. For the split of the thunderstorm, if there are more than one cluster within the analysis region, the cluster with the largest area is set as the main body of the thunderstorm. The cluster whose discharge centroid is more than 10 km away from the main cluster's discharge centroid will be seen as the split part of the thunderstorm and be discarded. Each time the window advances 12 minutes, the cluster is updated and also its centroid. If the distance from the previous cluster to the next cluster is less than 50 km, or the current cluster exhibits an overlap with the previous cluster, the two clusters are regarded as the same thunderstorm and recorded. As we focus on the transit thunderstorm in this region, the storm with a duration less than 60 min and the farthest distance in longitude less than 50 km is not considered as the aimed object of study. The thunderstorm started with the appearance of the valid area (>25 km$^2$), while the ending time of the thunderstorm is when the cluster is less than 25 km$^2$ and can not be depicted by the algorithm any longer.

Based on the selected thunderstorm and the coordinate of the discharge centroid in each interval, we can obtain the distance that the thunderstorm runs and the direction it moves. The true North is set as the benchmark to illustrate the direction of the storm. The azimuthal angle equation is as follows:

$$Direction = \arctan \frac{C_{lon2} - C_{lon1}}{C_{lat2} - C_{lat1}} \quad (0° < Direction < 360°) \tag{4}$$

where subscripts 1 and 2 represent the discharge centroids of storm clusters at two consecutive time intervals.

Velocities of the cluster, determined by distances traveled in a 12-minute interval, are recorded as well (seen in Eq. (5)). The farthest distances (FD) that the thunderstorm moves in longitude and latitude during the lifetime help to foresee the movement of the storm, the expressions of which are shown in Eq. (6) and Eq. (7).

$$Velocity = \frac{\sqrt{(C_{lat2} - C_{lat1})^2 - (C_{lon2} - C_{lon1})^2}}{12} * 60 \tag{5}$$

$$FD_{lon} = \max(C_{lon}) - \min(C_{lon}) \tag{6}$$

$$FD_{lat} = \max(C_{lat}) - \min(C_{lat}) \tag{7}$$

where the unit of velocity is km h$^{-1}$.

To characterize the motion of thunderstorms, we use five parameters mentioned above to depict their movements, which include the valid area, velocity, direction, and farthest distance in longitude and latitude. The meaning of the parameters is shown in Table 1.

## 3 Result

### 3.1 Total lightning characteristics and temporal evolution of thunderstorm

From the thunderstorm activities detected by FTLLS in the summer of 2014, eight thunderstorms around the Pearl River Delta region are selected which were evenly distributed in the morning, midday, afternoon and evening from 17 May to 23 May. Table 2 provides the basic information about these thunderstorms, including the date, the specific time, the duration, the total number of lightning events, lightning event rate (the number of lightning events per hour) and event-type classification. A common trend of thunderstorms over the PRD region is that convection occurs most frequently during the afternoon due to solar heating (Chen et al., 2014; Chen et al., 2015). The life cycle durations range from 2 hours to 4.2 hours, with a large difference in the number of total lightning.

The thunderstorm with the highest number of total lightning occurred on the afternoon of 23 May, consisting of 101,242 lightning events within 3.6 hours. The lowest number of total lightning events occurred on the evening of 19 May, lasting for 2.4 hours with 14,926 lightning events in total. The midday and afternoon thunderstorms keep relatively strong and stable, with more than twelve thousand lightning events per hour, while the morning thunderstorms are much more gentle and weaker, with around ten thousand lightning events per hour. The two evening thunderstorms are much variable and differentiated, with the first case possessing the highest frequency (29,240) per hour and the second case possessing the lowest frequency (7,263) per hour.

In the LLS data set of all storms, IC events make up 81.7% of total lightning and CG events make up 17.5% of total lightning. The NBEs make up the very slightest proportion, with less than 1% of total lightning. The 17 May afternoon storm consisted of 81,872 lightning events, with IC events accounting for the highest proportion (91.8%) among all storms. The distinctly high proportion of IC events occurred on the 23 May afternoon storm and the 17 May evening storm, which are stronger storms than other cases, indicating that a high IC ratio is in connection with severe weather. However, the weak 21 May morning storm also possesses a high proportion of IC events (88.7%), while other storms with a similar scale show a much lower IC ratio. Overall, the proportion of IC events is variable in thunderstorms, making it difficult to use them as a tool for predicting severe weather.

General total lightning distributions with respect to the time of eight cases are presented in Figure 4. The comparison between lightning events detected by the FTLLS (green shaded areas) and the chosen thunderstorms (light blue shaded areas) shows that there are some other storms dispersed within the LLS detection area. The less intense storms are excluded by the thresholds and the most prominent storms are selected chosen to characterize the movements. Note that the light blue shaded area in each statistical time interval was derived within the thunderstorm system, the analysis period was defined from the starting time to the ending time of thunderstorms, when the main body of the thunderstorms is well observed by radar. The blue line and red lines represent the IC events and CG events produced by the chosen thunderstorm, respectively. We can see that the variation of IC events is highly consistent with the total lightning events, while the variation of CG events is quite different. The thunderstorm occurred at 18:00 on 17 May and produced the largest number of total lightning events per 12 minutes, with the number being more than 8000 times. Another night storm that occurred at 19:12 on 19 May is much weaker, whose scale is slightly smaller than the morning that occurred at 8:36 on 20 May, with a smaller peak of total lightning per hour and a smaller number of total lightning events.

## 3.2 Spatial footprint of thunderstorms

The footprint, trajectory, and lightning event density of thunderstorms are displayed in Figure 5. The storm footprint is defined as the combination of the unique area consisting of the VA of each cluster and the path of the centroid. During the lifetime of all thunderstorms, the horizontal movement of the thunderstorm does not exceed 150 km in longitude and 100 km in latitude, except that the path of the 18 May afternoon thunderstorm is longer than the average and depicted in the 200 km×150 km domain.

The coverage and intensity of thunderstorms during the whole process can be visually presented in the evolution map. At the initiation stage, the VA is much smaller than that in the development or maturation stage when the thunderstorm moves faster in the meanwhile. Note that there is some interspace between two centroids when the storms move fast and the VA is not big enough. The reason is that the circle on the map is the equivalent of the valid area and can only reflect the value of the thunderstorm area within 12 min.

The transition of storms is mainly from west to east with long tracks, usually initiating in the Foshan district, crossing the Guangzhou district and disappearing in the Dongguan district. However, the 17 May midday storm is inclined to twist and spin in the same place with extremely short tracks. The VA of the 19 May evening storm is relatively small in each time interval compared with other storms, which is in accord with the small number of total lightning analyzed above, however, the velocity is no slower than any other severe storms.

**3.3 Duration time, valid area, movement velocity, farthest distance and direction**

     Figure 6 displays two lightning parameters to characterize the intensity and movement of thunderstorms: valid area and velocity, which can comprehensively show the track of thunderstorms in a measurable way. Despite the different periods of a day, there are two kinds of distribution of thunderstorm valid area in the whole evolution process. The first distribution is characterized as one-peak seen in Figure 6(a)(e)(g), which means the variation of VA rises at first and drops dramatically. The
225 valid area shows an upward trend at the beginning and decreases at last. The rising period is found to be longer than the drop period, which means that the peak of VA lies in the mature stage of thunderstorms. The second distribution is defined as the two-peak distribution in Figure 6 (b)(f)(h), which means there is a distinct decrease between two peaks during the lifetime of the thunderstorm. The VA increases in the initiation stage and reaches the first peak in the developing stage. After a small decline, the VA surges in a short time and arrives at the second higher peak in the mature stage. At last, the VA decreases
rapidly as the thunderstorm is dissipating. Figure 6(c)(d) is not in full accord with the one-peak distribution but almost close to it. Although there is another much smaller peak in the dissipating stage, it can be seen as the normal fluctuation. The difference is that the VA shows a slight increase sign after the highest peak. Meanwhile, the peak time of the 17 May midday storm is much earlier than the typical one-peak distribution, of which the peak time basically occurs in the mature stage. It is noticed that the VA does not decrease to zero at last because of the existence of time interval and the threshold of cluster area.
The velocity exhibits more marked changes with time. It oscillates severely compared with the valid area which shows a steady increase or decrease during the lifetime of thunderstorms. There is no stable variation in the velocity of thunderstorms, indicating that the instability of convection within the cloud. When the number of lightning events grows up and the valid area becomes bigger during the development of thunderstorms, the velocity does not show an obvious increase tendency. This finding is of great significance to facilitate our knowledge of the kinematics of the mesoscale convection system. The physical
mechanism inside the convective cloud needs further study in the future.

     The violin plots in Figure 7 present the cluster attributes including the VA and velocity in the whole thunderstorm process. The VA boxes in the afternoon show the rugby-shaped distributions, indicating that the storms are variable during the life cycle and extremely severe in the mature stage (seen in Figure 6 (e)(f)). The storms in the morning, midday and evening are much stable and the VA shows a uniform distribution, exhibiting the rectangle-shaped boxes. There is a great difference
between the two evening storms, with both the biggest and smallest maximum in this period, which is because of the instability of convection in this region. A storm with a big VA does not mean a fast speed when it moves. The velocity of each storm is

densely distributed around the median, with only one value much bigger than others, showing the rugby-shaped distributions. Although the VA of 17 May evening storm is much bigger than another evening storm, the speed does not greatly discriminate between each other.

Five characterization parameters are listed in Table 3 to reveal the kinematic features of thunderstorms. The maximum VA represents the biggest coverage that the thunderstorm affects within a 12-minute interval. The afternoon storm is notably more severe and intense than that in the morning and midday, while the two evening storms distinguish greatly from each other. The maximum VA is 891 km$^2$ occurred on the evening of 17 May, with the mean value being 662.7 km$^2$. However, the storm with minimum VA also occurs on the evening of 19 May, the maximum and mean of which are 253 km$^2$ and 146.7 km$^2$, respectively. It can be found that the velocity does not match precisely with the VA. The storm with the highest speed occurred on the morning of 20 May, with a value of 170.7 km h$^{-1}$. The lowest maximum of speed was 96.1 km h$^{-1}$ occurred on the evening of 19 May. Although total lightning events of these two storms are the smallest among eight cases, the speed of the cluster does not show the same characteristic.

To measure the horizontal motion of thunderstorms during the whole process, the horizontal farthest distances (FD) in longitude and latitude are calculated by the coordinates of centroids. It can be clearly seen that the longitudinal FD is much longer than the latitudinal FD, which means that the movement of the storm is mainly along the east-west path. The maximum and the minimum FD in longitude are 153 km and 55 km, respectively. The FD in latitude is much shorter than that in longitude, with the maximum and minimum being 45 km and 12 km.

Figure 8 illustrates the direction of the cluster. We gathered the direction of all cases in the normalized timeline to show the orientation. The motion of storms shows a distinct pattern, as the spread of direction distributes tightly in the range of 0°-180°. Combining with the ground track of thunderstorms in Figure 7, we can clearly see that the storms initiate in the west of the PRD region, moving to the east and disappearing after the thunderstorms travel around 106.5 km. This kinetic information could shed light on further research on severe convection weather prediction.

## 4 Discussion and Conclusion

For the purpose of characterizing the dynamic movement of thunderstorms as well as the associated attributes of lightning clusters over the PRD region, we investigate eight cases that occurred within a week from 17 May to 23 May in 2014. Based on the high-resolution total lightning data set obtained from VLF/LF Foshan Total Lightning Location System, the temporal and spatial characteristics of thunderstorms are presented in this study. To analyze the thunderstorm cluster features, statistics of various cluster parameters have been calculated.

Using the 8-adjacent connected-neighborhood labeling algorithm, five parameters are put forward to measure thunderstorm kinematics features, including the duration time, valid area, movement velocity, movement direction and farthest distance in longitude and latitude. Table 4 shows the comparison of the parameters between eight cases in the PRD region and previous

studies. Various thresholds are set to better capture the movement of thunderstorms. Miller and Mote (2017)identified the thunderstorm as the region of contiguous radar reflectivity greater than or equal to 40 dBz using connected neighborhoods

labeling. A sophisticated approach was taken to identify the rainfall pixels by Rigo et al. (2010)who used the radar reflectivity, pixel area and duration as the thresholds. A lower reflectivity threshold of 12 dBz was chosen to ensure all possible clusters, while the threshold of other parameters can appropriately avoid non-precipitation echoes in the rainstorm. In this paper, the valid box less than one event, the area of clusters less than 25 km$^2$ and the duration of the storm less than 60 min are neglected to reduce the number of small ground clusters and track the main storms within the analysis region.

Eight thunderstorms, which were evenly distributed in the morning, midday, afternoon and evening, are selected to compare the different kinematic features from 17 May to 23 May 2014. Most previous studies focus on the interannual or interdecadal variations in the characteristics of the storms. Kohn et al. (2011) selected 670 winter storms and 13,600 summer storms in 2008 to track the spatial and temporal attributes over the Mediterranean area and Europe. Harrison and Karstens (2017) climatologically analyze the fundamental components of thunderstorm geospatial movements within the continental United

States. This paper is aimed to make a comprehensive analysis of the cases in a week and reveal the dynamic motion of thunderstorms over the PRD region in the south of China.

Significant characterization parameters are proposed as metrics to depict the kinematic features of thunderstorms, including the duration, VA, velocity, direction, and FD in longitude and latitude during the evolution of thunderstorms. It is found that no more than three thunderstorm parameters are demonstrated in the previous study. Rigo et al. (2010) reported the

295 duration and the average area of 66 Catalonia warm-season thunderstorms. The lifetime was between 54 minutes to approximately 8 hours, with the average duration of the whole thunderstorm evolution process being about 3.5 h, which is slightly longer than this study (2.93 hours). The average area of 66 thunderstorms was 509 km$^2$ in a 6-minute time interval, with the biggest cluster area in the mature stage. A June supercell propagated north of Munich in the eastern direction was reported by Meyer et al. (2013a) to illustrate the area, velocity and farthest distance of storms, showing that the maximum cell

area was nearly 500 km$^2$ in the 3-min interval. The average area of eight thunderstorms in this paper is 336 km$^2$ per 12 min. The differences derive from the geographic position, the severity of thunderstorms, the clustering methodology and so forth.

In this paper, there are two kinds of distribution to describe the variation of the valid area during the lifetime of thunderstorms, which are one-peak distribution with the maximum in the mature stage and two-peak distribution with a relatively smaller peak in the early time of the storm. The maximum VA is 891 km$^2$ occurred on the evening of 17 May, with

305 the mean value being 662.7 km$^2$. The storm with minimum VA also occurs in the evening, the maximum and mean of which are 253 km$^2$ and 146.7 km$^2$, respectively. The area variation of Meyer's case occurred on 25 June 2008 appears to be more fluctuant, with a sharp decrease in the developing stage and many peaks during the whole evolution process (Meyer et al., 2013a). The maximum of cluster areas varies notably between storms reported by Betz et al. (2008), the largest area reaches up to 7000 km$^2$ in 10 min, while the smallest area is only 1550 km$^2$. The intensity of storms discriminates in different periods

of a day in this paper, but not in a big difference, indicating that the convection in the summer season is severe but stable in

this region. To be noted, the valid area observed by the FTLLS is much smaller than that observed by the radar, where the former represents the lightning discharge activity and electricity charge accumulation, and the latter reflects the content of hydrometeors and the effect on radar echoes (Xu et al., 2010; Miller and Mote, 2017).

The velocity of thunderstorms obtained by the motion of lightning centroids in this paper represents the integral movement which is basically composed of three types of factors: the translation (synoptic), the forced propagation (mesoscale) and the autopropagation (thunderstorm itself) (Cotton et al., 2011). In this paper, velocities are calculated by the discharge centroids of the thunderstorm. Owing to the instability of updraft and non-inductive electrification in the convective cloud, the discharge centroid is not always the barycentre of thunderstorm clusters. As the movement metrics are obtained through the lightning events, which represent the electrification in the cloud, the discharge centroid can better reflect the electrification variations in the storm. The storm with the highest speed occurred on the morning of 20 May, with a value of 170.7 km/h. The lowest maximum speed was 96.1 km/h occurred on the evening of 19 May. The velocity does not show the same tendency as the variation of VA during the lifetime of thunderstorms. It oscillates severely compared with the valid area which shows a steady increase or decrease during the lifetime of thunderstorms. The fluctuation of velocity is very likely to result from the calculation method of centroid discharge. In addition, the different time intervals may cause the bais in velocity. Some severe thunderstorms like supercells last for a short time and move extremely fast, leading to poor monitoring results. A relatively large velocity variation is also seen in the Mediterranean storm (Betz et al., 2008), but with a general upward trend in some cells during the whole movement. Meyer et al. (2013a) proposed that long-lived storms are most likely fast propagation as the storms with velocities around 80 km/h spent 150 min to 240 min to cross the domain, however, this was under-represented because of the insufficient statistics. The eight cases in this study also do not show this trend.

Affected by the South China Sea Summer Monsoon, the motion of storms shows a distinct pattern, as the spread of direction distributes tightly in the range of 0°- 180°, indicating that thunderstorms mainly move from west to east. The orientation of the thunderstorm can be affected by the topographic relief (Miller and Mote, 2017). Lin et al. (2011) found that the warm season afternoon thunderstorm over Taiwan Island frequently occurred in a narrow strip, parallel to the orientation of the mountains, along the lower slopes of the mountains. The urban heat island effects and northern mountains in Guangzhou city may have an influence on the movement of thunderstorms over the PRD region (Yin et al., 2020).

Overall, the detailed analysis of the dynamic movement of eight thunderstorms in May shows that there are some remarkable characteristics, but still exist variations among thunderstorms in different periods of a day over the PRD region in the summer season. The result helps to improve the recognition of severe thunderstorms in advance by giving a general understanding of how long the storm lasts, how fast the cluster moves and how much area the storm affects, via information about the kinematics features of thunderstorms, and ideally establish a foundation for future research that may contribute to the development of a new or improved prediction paradigm.

Code and data availability. The data archiving is underway and will be presented at the Zenodo repository. The data have been temporarily
uploaded as Supporting Information for review purposes.

Author contributions. JWand LC lead the lightning observation program. JW, LC, SC and QL participated in the establishment of the Foshan
total lightning system. SC contributed to the conceptualization of the study. JW and SC wrote the manuscript. YF, MZ, YH and RS revised
and improved the text. All authors have read and agreed to the published version of the manuscript.

Competing interests. The contact author has declared that neither they nor their co-authors have any competing interests.

Acknowledgements. The authors express their gratitude to all the members for their contribution to the lightning detection system.

Financial support. This work was supported by the National Natural Science Foundation of China (Grant Number: 51807144).

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

**Figure**


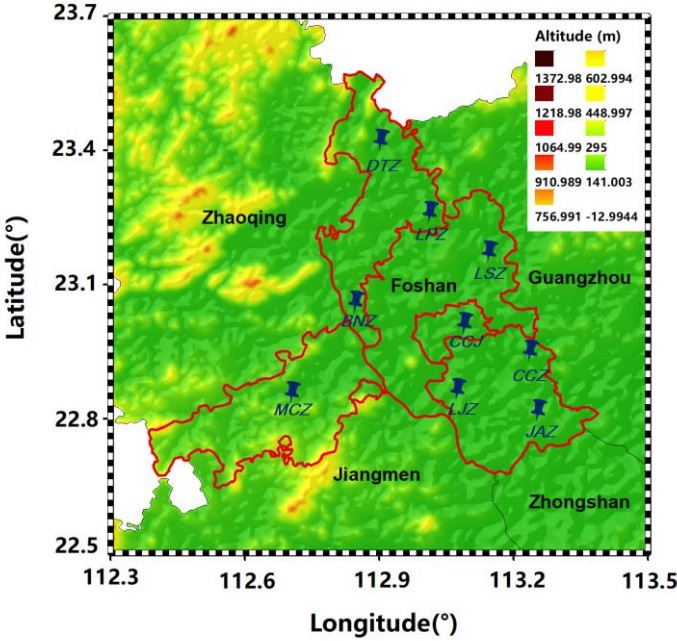

**Figure 1. Geographical distribution of the Foshan Total Lightning Location System (FTLLS), in which a full operation of nine stations in the Foshan area is shown. Station location is displayed by blue icons.**


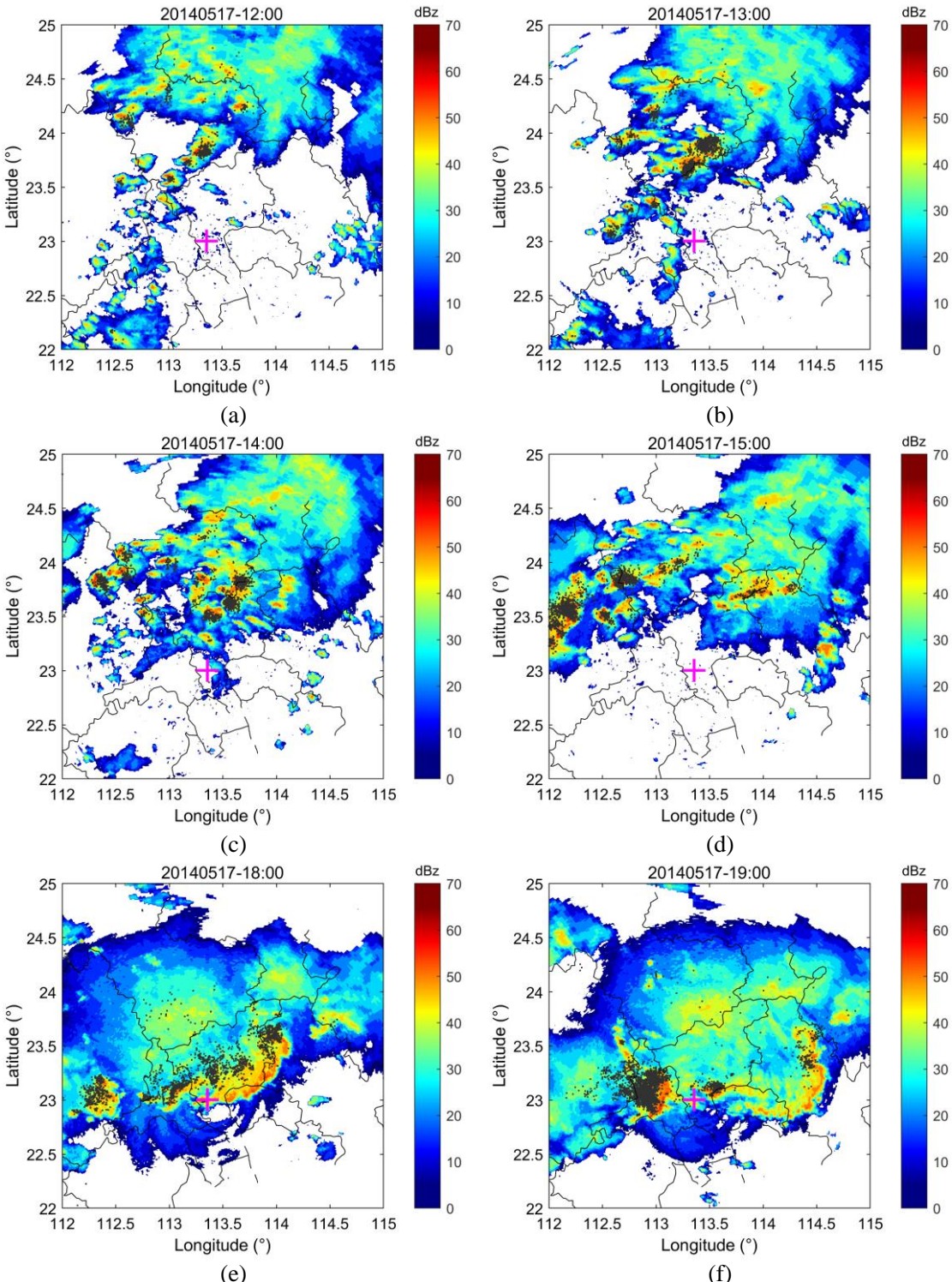

(a)

(b)

(c)

(d)

(e)

(f)

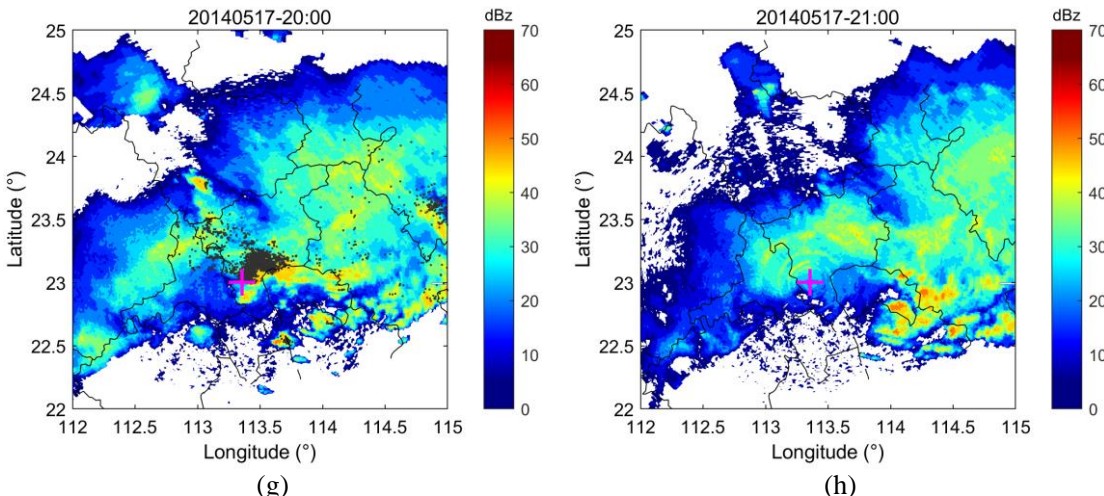

**Figure 2. The stacking map of reflectivity scan by WSR-98D research radar and spatial distributions of total lightning data on 17 May 2014. Prominent areas of higher reflectivity are simultaneously covered by lightning events. The red plus represents the radar station which is situated on the east side of the FTLLS.**

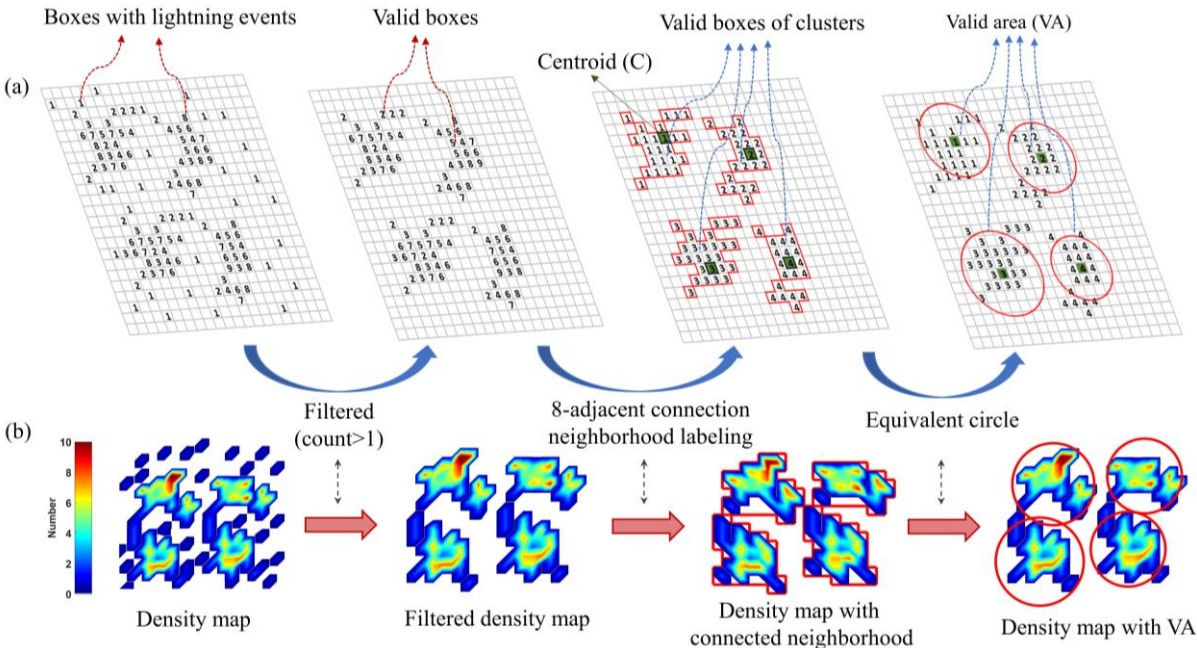

**Figure 3. (a) Illustration of the 8-adjacent connected-neighbourhood labeling and the procedure of dealing with the lightning data to obtain the centroid and the valid area of clusters. (b) The workflow of cluster identification through density map.**

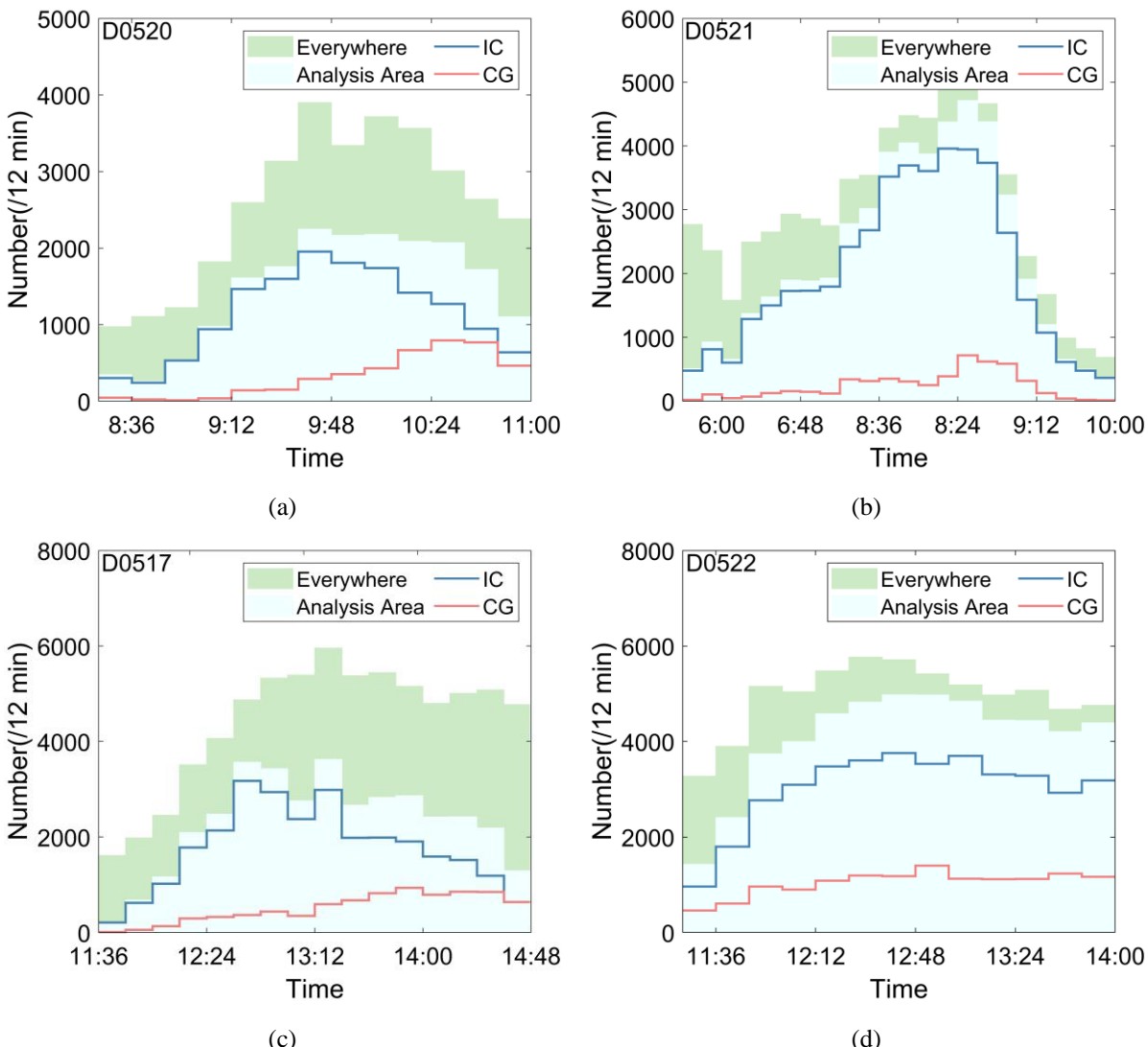

(a)

(b)

(c)

(d)

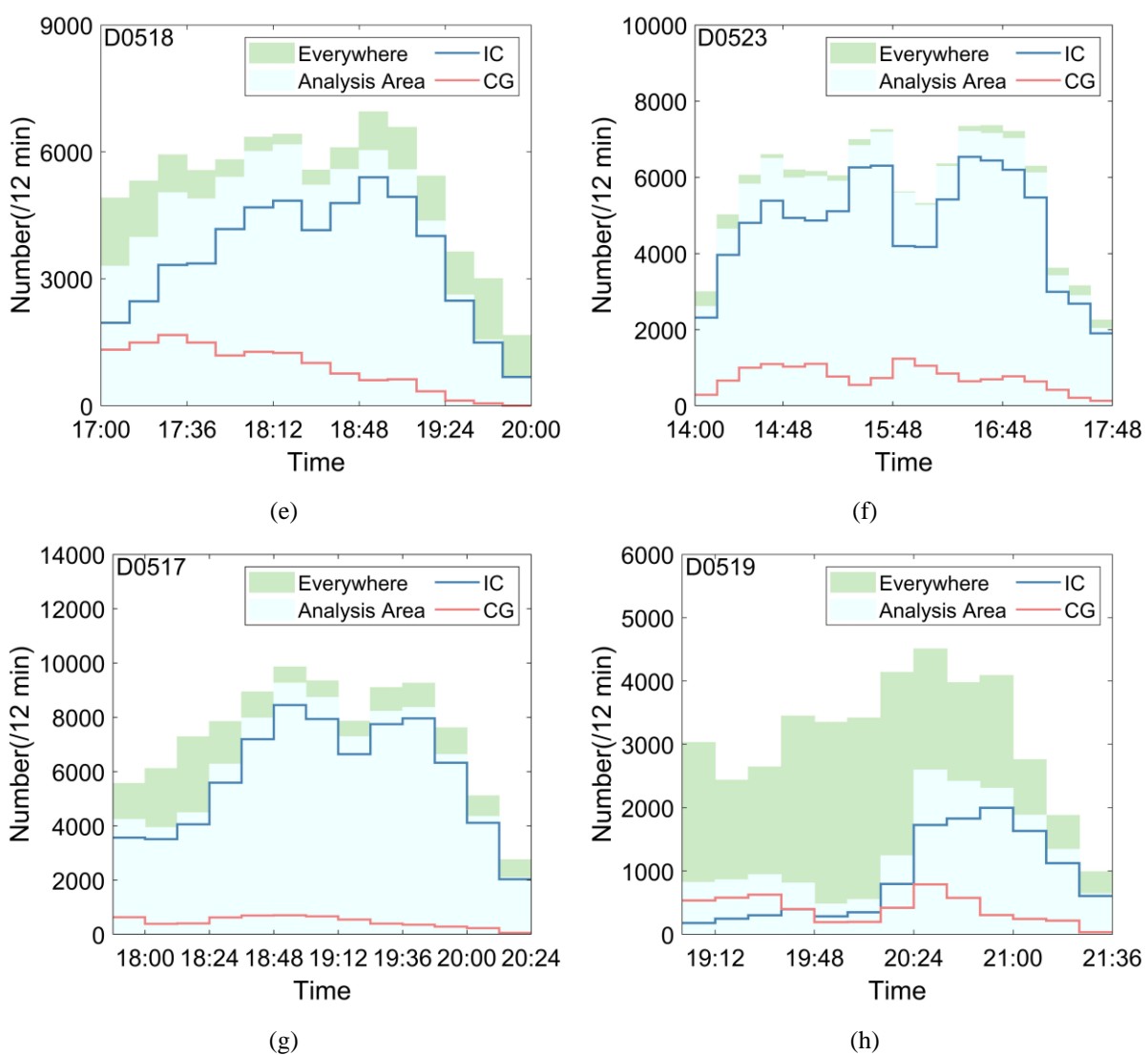

(e)    (f)

(g)    (h)

**Figure 4. (a-h) Lifetime diagrams of total lightning events for thunderstorm case study from 17 May to 23 May. The green shaded areas represent all total lightning events detected by the LLS network, whereas the light blue shaded areas represent total lightning events produced by the chosen thunderstorm. The blue and red lines indicate the number of IC events and CG events with respect to time per 12 min, respectively.**



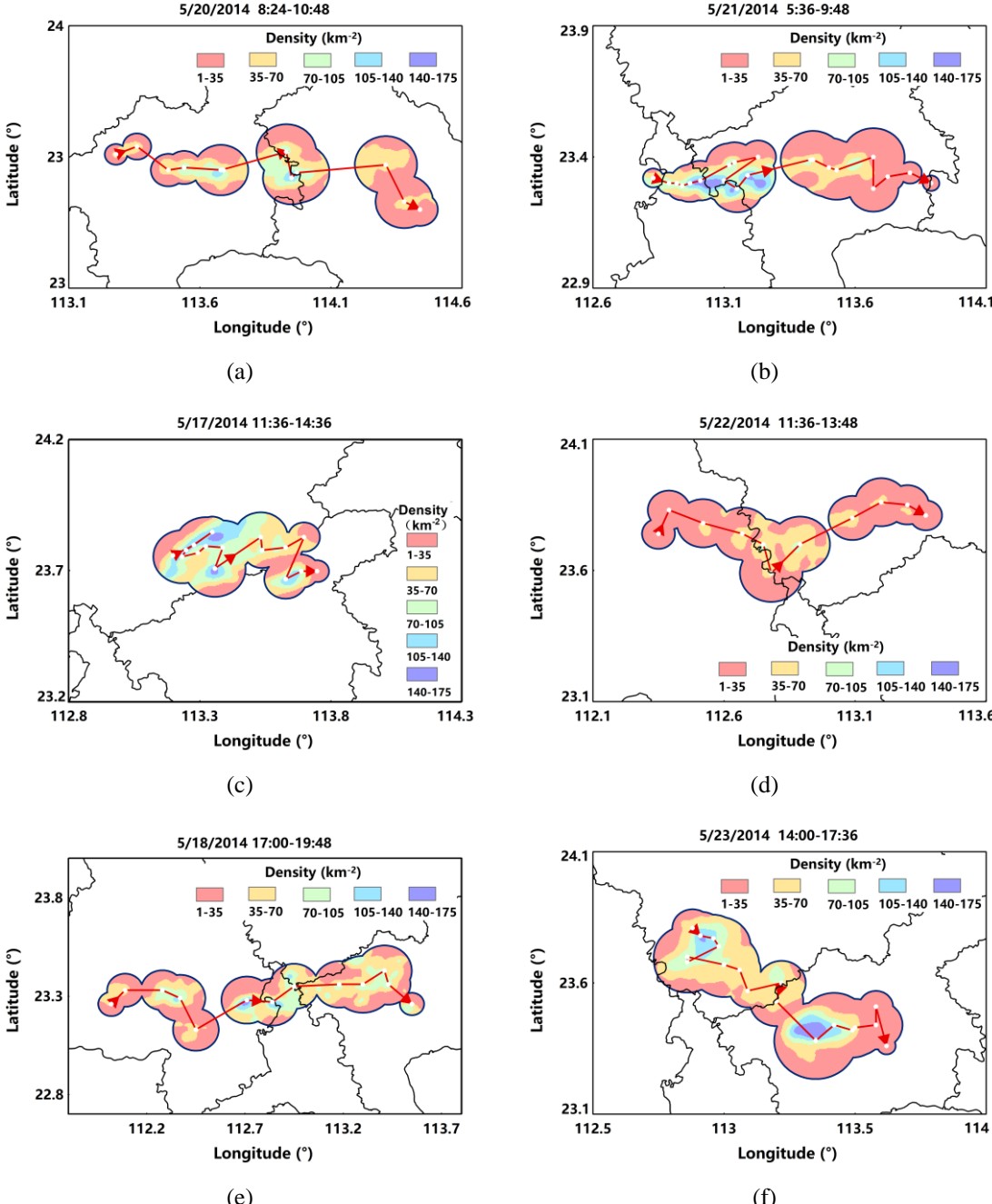

(a)

(b)

(c)

(d)

(e)

(f)

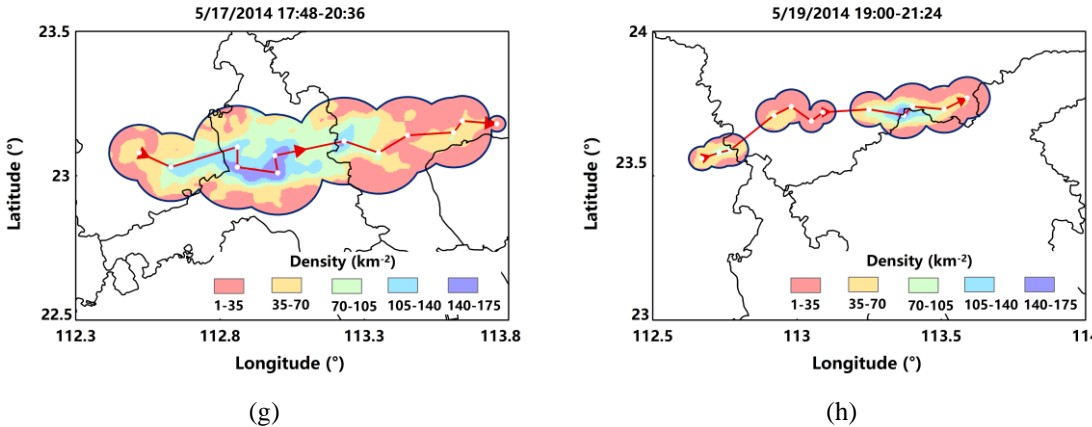

(g)                                                                                    (h)

**Figure 5. Ground tracks of eight thunderstorms occurred over the Pearl River Delta region in May, 2014. The horizontal axis corresponds to the longitude with the vertical axis standing for the latitude. The white solid dot is the discharge centroid of the valid area in 12 min intervals. The solid red line between two dots is the general path of the thunderstorm and the VA that the storms cover is expressed by the centroid-centered circle of a time interval. The lightning density of thunderstorms during the whole process is shown within the valid area.**



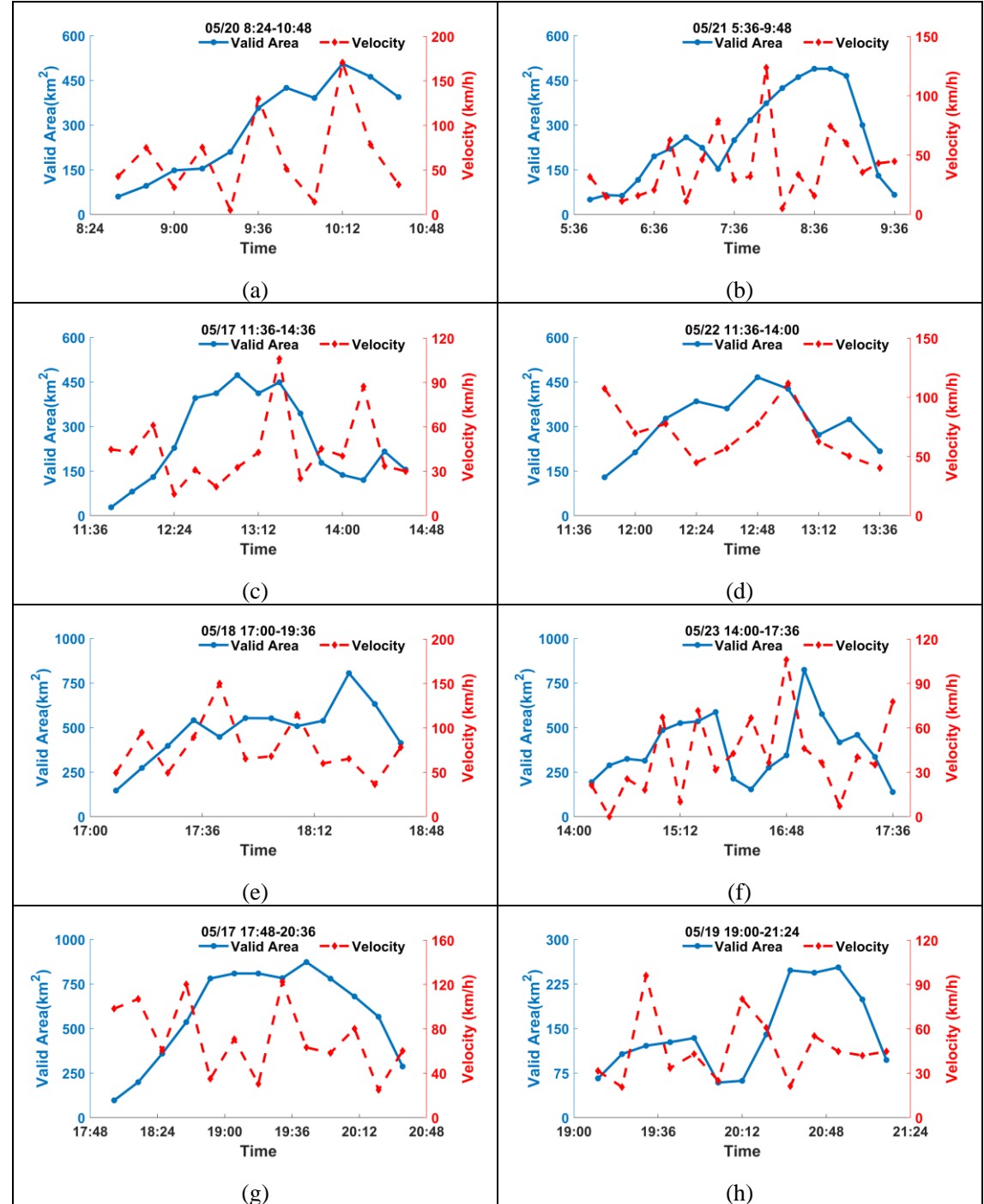

**Figure 6. Valid area and velocity of eight cases, as a function of time. The blue solid line and the red dotted line represent**

**the valid area and velocity within a time interval, respectively.**

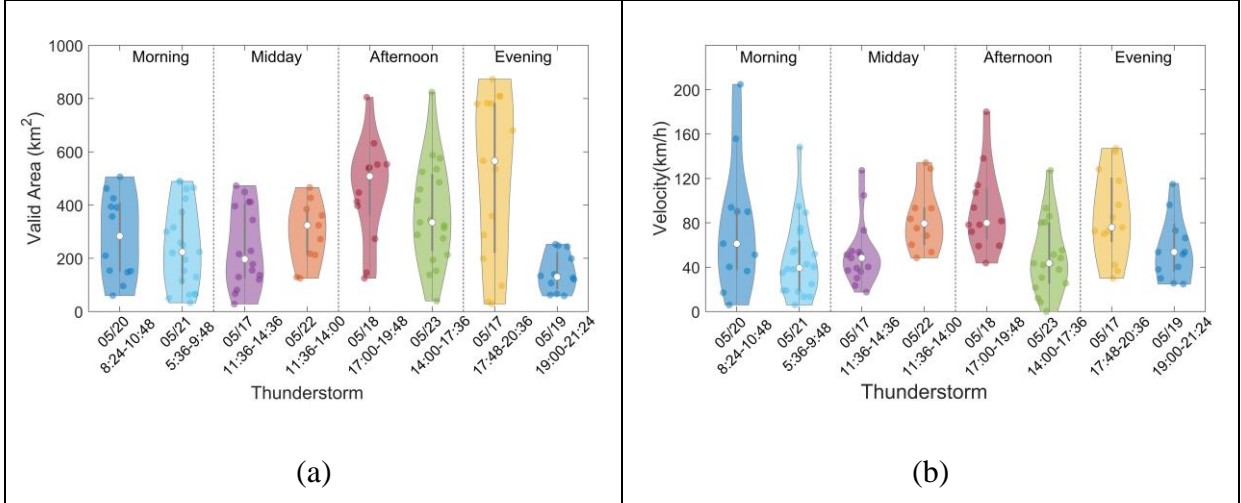

**Figure 7. (a) Comparison of the valid area between eight cases. The upper and lower edges of boxes mean the maximum and minimum values. The while circle in each box represents the median of VA. The filled circles with a darker color than the box represent all VA values of thunderstorms. The grey vertical line in the box represents the interquartile range of VA. (b) Same as (a), but for the velocity.**

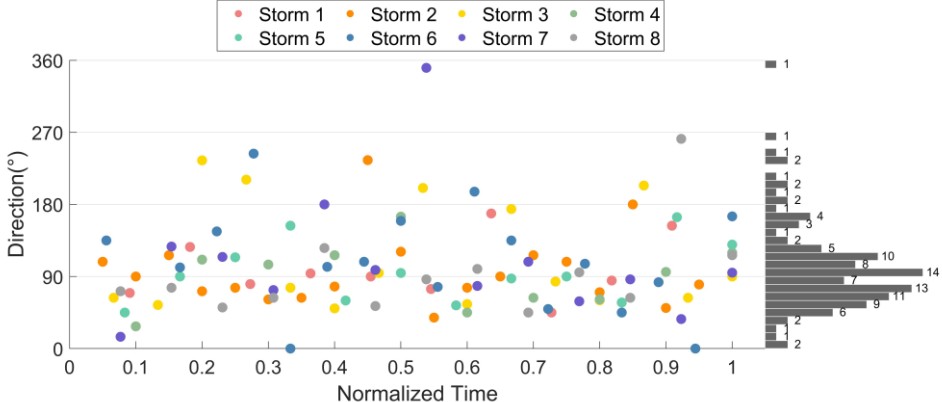

**Figure 8. The cluster direction of motion in each interval. The colored dots denote different thunderstorms. As the duration of thunderstorms is different, the normalized time is used for the gathering of all directions of thunderstorm movements. The bar chart on the right represents the frequency of the 10° angle range.**

**Table 1. Five parameters to characterize the motion of thunderstorms**

| Parameter | Indication |
|---|---|
| Duration time (h) | The thunderstorm duration of the whole evolution process |
| Valid area ($km^2$) | The area that the thunderstorm affects, obtained by the number of valid boxes. |
| Movement Velocity ($km\ h^{-1}$) | The speed between two centroids. |
| Movement Direction (°) | The angle of motion, taken the east-west direction as a benchmark. |
| $FD_{lon}$ (km) | The farthest longitudinal distance of centroids during the lifetime of a thunderstorm |
| $FD_{lat}$ (km) | The farthest latitudinal distance of centroids during the lifetime of a thunderstorm |


**Table 2. Overall characteristics of eight thunderstorms occurred within a week**

| | Storm | Date | Time (LT) | Total lightning | | IC events | | CG events | | NBEs | |
|---|---|---|---|---|---|---|---|---|---|---|---|
| | | | | All | No.(hour$^{-1}$)[a] | No. | %[b] | No. | %[b] | No. | %[b] |
| Morning | 1 | 05/20 | 8:24-10:48 | 17,985 | 8,175 | 14,215 | 79.0 | 3,725 | 20.7 | 45 | 0.3 |
| | 2 | 05/21 | 5:36-9:48 | 46,681 | 11,115 | 41,411 | 88.7 | 4,945 | 10.6 | 325 | 0.7 |
| Midday | 3 | 05/17 | 11:36-14:36 | 37,418 | 12,473 | 28,325 | 75.7 | 8,519 | 22.8 | 574 | 1.5 |
| | 4 | 05/22 | 11:36-14:00 | 44,694 | 18,623 | 33,305 | 74.5 | 11,168 | 25.0 | 221 | 0.5 |
| Afternoon | 5 | 05/18 | 17:00-19:48 | 65,786 | 23,495 | 52,126 | 79.2 | 13,250 | 20.1 | 410 | 0.6 |
| | 6 | 05/23 | 14:00-17:36 | 101,242 | 28,123 | 87,041 | 86.0 | 13,558 | 13.4 | 643 | 0.6 |
| Evening | 7 | 05/17 | 17:48-20:36 | 81,872 | 29,240 | 75,118 | 91.8 | 6,028 | 7.4 | 726 | 0.9 |
| | 8 | 05/19 | 19:00-21:24 | 17,433 | 7,263 | 11,780 | 78.9 | 3,007 | 20.1 | 139 | 0.9 |
| Average | | | | 51,326 | 17,278 | 42,915 | 81.7 | 8,025 | 17.5 | 385 | 0.8 |

a. The number of total lightning events per hour.

b. The ratio of IC events to total lightning events.


**Table 3. Parameters to characterize thunderstorms of eight cases.**

| | Date | Duration (h) | Valid Area (km$^2$/12 min) | | Velocity (km h$^{-1}$) | | Longitudinal Distance (km) | Latitudinal Distance (km) |
|---|---|---|---|---|---|---|---|---|
| | | | Max | Mean | Max | Mean | | |
| Morning | 05/20 | 2.2 | 506 | 279.6 | 170.7 | 64.2 | 116 | 24 |
| | 05/21 | 4.2 | 489 | 244.8 | 123.7 | 39.5 | 106 | 12 |
| Midday | 05/17 | 3 | 473 | 239.1 | 106.1 | 43.8 | 55 | 18 |
| | 05/22 | 2.4 | 466 | 295.1 | 111.8 | 69.9 | 108 | 27 |
| Afternoon | 05/18 | 2.8 | 805 | 456.2 | 150.1 | 76.8 | 153 | 30 |
| | 05/23 | 3.6 | 824 | 369.8 | 106.1 | 41.1 | 76 | 45 |
| Evening | 05/17 | 2.8 | 891 | 662.7 | 122.6 | 60.6 | 146 | 20 |
| | 05/19 | 2.4 | 253 | 146.7 | 96.1 | 46.6 | 92 | 21 |
| Average | | 2.93 | 588.38 | 336.75 | 123.4 | 55.3 | 106.5 | 24.63 |

**Table 4 Parameters comparison between the summer thunderstorms in the PRD region and previous studies.**

| | Data | Threshold | Cases | Duration (h) | Area (km$^2$) | Velocity (km h$^{-1}$) | Farthest Distance (km) | Direction (°) |
|---|---|---|---|---|---|---|---|---|
| This paper | FTLLS | Area > 25 km$^2$ Duration >60 min | 8 cases | 2.93 | Mean: 336.75 (in 12 min) | Max:123.4 Mean: 55.3 | Longitudinal: 106.5 Latitudinal: 24.63 | 270-360 |
| Betz et al. (2008) | LINET | / | 19 June 2007 Cell#8 | / | Max: nealy 1550 (in 10 min) | Max: nealy 90 | / | / |
| | | | 19 June 2007 Cell#3 | | / | Max: nealy 80 | | |
| | | | 21 July 2007 Cell#29 | | Max: nealy 7000 | Max: nealy 60 | | |
| | | | 21 July 2007 Cell#51 | | Max: nealy 2500 | Max: nealy 55 | | |
| Rigo et al. (2010) | LINET & Radar | >12 dBz Area >24 km$^2$ Duration >50 min | 66 cases | 3.5 | Mean: 509 (in 6 min) | / | / | / |
| Kohn et al. (2011) | ZEUS | | Summer (13,600 cases) | / | Max: nealy 3400 (in 15 min) | Mean: 16.5 | / | / |
| | | | Winter (670 cases) | / | Max: nealy 3700 | Mean: 6.2 | | |
| Meyer et al. (2013b) | LINET & Radar | Duration >35 min | 12 May 2008 | / | | 80 | Diagonal length: 28 | / |
| | | | 25 June 2008 | / | Max: nealy 500 (in 15 min) | / | / | |
| Harrison and Karstens (2017) | / | / | 9 years in CONUS | / | / | Mean: 45 | / | 244 |
| Miller and Mote (2017) | Radar | >40 dBz Duration >30 min | Pulse thunderstorm (5378 cases) | 2.95 | Medium: 470 (in 5 min) | / | / | / |
| | | | Weakly forced thunderstorm (885,496 cases) | 0.78 | Medium: 42 (in 5 min) | | | |
