# Peer review of "Characterizing the dynamic movement of thunderstorms using VLF/LF total lightning data over the Pearl River Delta region"

_Atmospheric Chemistry and Physics, 2021_

## Author Comment (AC1)

**Referee Comment 1**

**General comments**

The work reveals the characteristics of thunderstorm dynamic movement based on eight thunderstorms evenly distributed around the Pearl River Delta (PRD) region in southern China from May 17 to May 23, 2014. They found that the storms initiated in the west of the PRD region, then moved to the east and disappeared after the thunderstorm travels around 106.5 km in longitude. In addition, there are two kinds of distribution to depict the property of valid area, which are one-peak distribution and two-peak distribution. This manuscript used five parameters and found, affected by rivers in the Pearl River Delta region, the motion of storms shows a distinct pattern, as the spread of direction distributes tightly in the range of 0°- 90° and 270°-360°. The overall structure and layout of the manuscript is clear and the experimental design is reasonable. I will suggest it to be accepted after addressing my comments below.

Authors' Response

We thank the referee for your encouraging comments

**Specific comments**

L189: The definitions of one-peak distribution and two-peak distribution in the manuscript are vague, and more detail is suggested.

Authors' Response

The authors would like to express our appreciation for the reviewer's suggestions, The one-peak distribution means the variation of VA rises at first and drops dramatically. The two-peak distribution means there is a distinct decrease between two peaks during the lifetime of the thunderstorm. We have added some explanations to make it more detailed.

L196: "Figure 5(c)(d) is not in full accord with one-peak distribution ... after the highest peak", is that means there may be three kinds of distribution of thunderstorm valid area in the whole evolution processes? Does it need to be considered separately?

Authors' Response

Thanks for your comment. The issue pointed out by the reviewer is very important. Figure 5 (c)(d) is not in full accord with one-peak distribution but very close to it. Although there is another much smaller peak in the dissipating stage, it can be seen as the normal fluctuation. The authors suggest that it can be considered as the one-peak distribution. The corresponding explanation has been added to the manuscript.

L202, 220: The finding that velocity does not match precisely with the VA, which is differ from common cognition about thunderstorms, is mentioned several times in the manuscript. How is the common cognition (example references) and what causes this?

Authors' Response

Thanks for your comments. The authors have looked up the literature and found that few papers mentioned the relationship between velocity and area of thunderstorms. We believe that this is a good innovation point of this article. As for the mechanism inside the convective cloud, we believe that more research can be done to explain this finding

in the future. We will continue to study the movements of thunderstorms and the microphysical principle inside the storm. The corresponding explanation has been added to the manuscript.

**Typing errors**

L8, 250: "was" should be replaced by "were".

L142: The full name of "PRG" should be elaborated.

L187: "3.2" should be replaced by "3.3".

L241, 245: "duation" should be replaced by "duration".

L243: "setted" should be replaced by "set".

L247: "aviod" should be replaced by "avoid".

L274: "acitivity" should be replaced by "activity".

Authors' Response

Thanks for your comments. The typing errors have been checked and revised throughout the article.

---

## Author Comment (AC2)

**Referee Comment 2**

This manuscript employs total lightning observations from the Foshan total lightning locating system to characterize the movement and size of eight thunderstorms in 2014. However, it isn't clear how they are useful or novel. Additionally, I have concerns about the data processing leading to their conclusions. For instance, some of the storm velocities touted by the paper are physically unrealistic (e.g., >200 km h$^{-1}$). I recommend rejection in the present form.

Authors' Response

The authors thank the reviewer for providing all the suggestions and sincerely accept that these have turned out to be indispensable in pushing and improving the standard of the current work.

There are two innovation points of this article. Firstly, it is the total lightning data derived from the VLF/LF Lightning Location System (LLS) that we use to reveal the characteristics of thunderstorms in the PRD region. In the literature, the lightning and thunderstorm-related research for a region is usually based on the radar data, CG LLS, Very High Frequency (VHF) Lightning Mapping Array and the satellite (Chen et al. 2014; Chen et al. 2015; Liu et al. 2014; Xu et al. 2009; Zhang et al. 2014). There has not been comprehensive research about the characteristics of total lightning in the PRD region. Few literature has used the total lightning data to reflect the movement of thunderstorms. The VLF/LF total lightning data enriches data sources and provide a new perspective to study the thunderstorm in this region.

Secondly, the authors put forward five parameters, including the duration time, valid area, movement velocity, movement direction, and FD in longitude and latitude to reveal the kinematic features of thunderstorms in the region. The duration reflects how long the storm will last. The movement velocity helps to exhibit how fast the thunderstorm travels and predict how long it will take to move to other places. The movement direction represents where the thunderstorm will go. The valid area shows the range the thunderstorm will affect. The farthest distance shows how far the thunderstorm will go. The lifetime and spatial movement of thunderstorms in the region can be clearly depicted with these characteristic parameters, which are beneficial to the thunderstorm prediction.

We want to clarify that the velocities are calculated by the discharge centroids of the thunderstorm. Owning to the instability of updraft and non-inductive electrification in the convective cloud, the discharge centroid is not always the barycenter of thunderstorm clusters. In some cases, as the discharge center lies on the left of the clusters at first, the cluster moves forward after 12 minutes with the discharge center moving to the right. The velocity may be relatively larger compared with the velocity calculated by the movement of the barycenter. However, as the movement metrics are obtained through the lightning events, which represent the discharge in the cloud, the authors believe that the discharge centroid can better reflect the electrification variations in the storm. The corresponding discussion has been added to explain the high value of speed.

**Major comments:**

1. While the paper indicates eight storms were selected for analysis, there is no justification given for why these eight storms were retained and others discarded. With only eight storms this study is missing the sample size to make generalizable conclusions while simultaneously lacking the detail of a case study. There either needs to be a larger number of storms considered, or these eight storms must be analyzed in greater detail (i.e., compare to convective environmental conditions, synoptic wind fields, etc). For instance, it is hard to conduct a thunderstorm morphology study without characterizing the environmental conditions (e.g., CAPE, wind shear, etc) since those parameters are influential determinants of the size and longevity of convection.

Authors' Response

Thanks for your comments. We want to clarify that there are two thresholds and limitations of thunderstorms we choose. As has been explained in the article, 'To better capture the main spatial movement of the thunderstorm, clusters less than 25 km$^2$ and the duration of storm less than 60 min are removed based on the scale of thunderstorms in the PRD region.' The less strong thunderstorms are not considered in the article since it is the strong thunderstorms that pose great damage.

In the literature, some analyze only one or two storms specifically (Betz et al. 2008; Meyer et al. 2013; Strauss et al. 2013), the others analyze thousands of thunderstorms without details, such as the track or the area (Rigo et al. 2010). In this article, the authors analyze eight thunderstorms at different times of the day with different intensities in detail. It not only shows the thunderstorm diversities in the PRD region, such as thunderstorms with heavy precipitation on 17 May and 23 May and mild storms in other cases (The authors have added the precipitation in 24 h to show the intensity of thunderstorms in the article), but also retains the details of storm and represents some common characteristics, such as the durations and directions of storms. The authors believe that this is another creativity in the article.

In order to verify the validation and explain the rationality of using total lightning data to characterize the movement of the thunderstorm, the authors have added the stacking map of reflectivity scan by DSR research radar and spatial distributions of total lightning data on 17 May 2014. We can see that prominent areas of higher reflectivity are simultaneously covered by lightning events. The authors believe that clusters of lightning events and their variations can effectively illustrate the thunderstorm evolution. Lightning-related research can definitely enrich the study of thunderstorm morphology.

In view of the reviewer's comment, the following description and figures have been added to the revised version of the manuscript.

*During the monsoon period (in May on average) in the PRD region, the South China Sea Summer Monsoon (SCSSM) enhances the precipitation owing to the southwesterly monsoon flow, especially the southwesterly low-level jets, carrying abundant water vapor to South China (Chen and Luo, 2018; Bei et al., 2002). More than 50% of heavy rainfall events in South China occur in April–June period, in which precipitation is*

*primarily related to fronts and monsoon flows (Wu et al., 2011). Persistent heavy rainfall occurred from 17 May 2014 to 23 May 2014, especially in the central and eastern part of the PRD region. It was found that 62 automatic weather stations recorded heavy precipitation of more than 100 mm on 17 May, while Huizhou and Shenzhen stations recorded 24-h rainfall of 377. 9 mm and 274. 6 mm (Bingzhi Zheng, 2015). Severe thunderstorms occurred in the PRD region with a record-breaking 24-h rainfall of 477.4 mm starting from 2000 LST (local standard time = UTC + 8 h) 22 May during this heavy rainfall week. The hourly precipitation in Conghua station surpassed 60 mm at 1300 LST on 23 May (Xinyu Zhou, 2017; Zhongqing Liang, 2015). Figure. 2 showed general radar characteristics and lightning distributions on 17 May. The lightning events are located in the area with radar reflectivity higher than 30 dBz, which has been verified as the threshold on the south of China (Xu et al., 2010). Consecutive precipitations within one week include the extremely severe thunderstorms and the relatively mild thunderstorms which serve as great cases for comparison.*

[Figure]

[Figure]

*Figure. 2 The stacking map of reflectivity scan by WSR-98D research radar and spatial distributions of total lightning data on 17 May 2014. Prominent areas of higher reflectivity are simultaneously covered by lightning events. The red plus represents the radar station which is situated on the east side of the FTLLS.*

2. I am concerned about the method of calculating the thunderstorm's direction, velocity, and furthest distance parameters. There is no mention of how thunderstorms spanning multiple 12-min grids are joined into a single multi-grid thunderstorm. While connected neighborhoods labeling can be performed in three dimensions, the paper does not indicate this capability was utilized. Figure 4 shows that 12-min increments were joined, but this aspect of the methodology is important and not discussed at all. Without a clear method of joining multi-timestep storms, it is hard to account for storm splits and mergers that could easily sway the velocity, duration, and FD calculations that span multiple 12-min grids. In fact, the maximum velocity reported in the abstract of >200 km h$^{-1}$ (a physically unrealistic value), as well as the highly variable storm velocities in Figure 5, suggests the methodology is not joining thunderstorms across multiple 12-min frames effectively. While the storm velocities are touted as a finding with "great significance" in lines 200-205, I believe it is more likely a deficiency in the methodology.

Author Response

Thanks for your comments. We have to clarify that we have explained how the thunderstorm clusters are identified and selected. Firstly, lightning events are collected

in 12-minute time interval and put into the 0.01*0.01 grid. The grids with more than one event are retained as the valid grid. The area of each grid is 1 km². Then, using the 8-adjacent connected-neighborhood labeling method, we can obtain clusters of lightning events within the analysis area, including the discharge centroid and the valid area. Clusters less than 25 km² are removed based on the scale of thunderstorms in the PRD region. Two clusters whose discharge centroid is less than 10 km merge as one cluster. Afterwards, each time the window advances 12 minutes, the cluster is updated and also its centroid. If the distance from the previous cluster to the current cluster is less than 50 km, or the current cluster exhibits an overlap with the previous cluster, they can be regarded as the same thunderstorm. At last, we obtain the coordinates of all clusters in the area and we choose the thunderstorms whose lifetime is longer than 60 minutes. The tracks are defined by the sequence of centroid positions. The more detailed explanation has been added to the manuscript.

As we have explained above, velocities are calculated by the discharge centroids of the thunderstorm. Owning to the instability of updraft and non-inductive electrification in the convective cloud, the discharge centroid is not always the barycenter of thunderstorm clusters. In some cases, as the discharge center lies on the left of the clusters at first, the cluster moves forward after 12 minutes with the discharge center moving to the right. The velocity may be relatively larger compared with the velocity calculated by the movement of the barycenter. However, as the movement metrics are obtained through the lightning events, which represent the discharge in the cloud, the authors believe that the discharge centroid can better reflect the electrification variations in the storm. Therefore, the velocity we calculated is reasonable.

3. The detection efficiency of the FTLLS will vary with distance from the network. Storms that move into the periphery of the detection area will experience inconsistent detection efficiency and the calculation of the movement metrics will be biased at these ranges. For instance, according to the longitudes in Figure 4, the storm in pane (a) extends nearly an entire degree east of the FTLLS domain in Figure 1. The detection efficiency, particularly for IC flashes, must erode at this distance, and making the calculations of FD, VA, and velocity questionable. The effect of the FTLLS detection efficiency on the thunderstorm classification needs to be investigated.

Author Response
We want to clarify that FTLLS can locate the lightning radiation source within a distance of 100 km in the Foshan area accurately, from which the data provides valid reference information for analyzing lightning strike faults on power transmission lines (Cai et al. 2019b). Cai et al. (2019a) has reported that the average horizontal error within a distance of 100 km is less than 1.5 km and the corresponding vertical error is less than 1000 m (seen in Figure 9). The stacking map shows that the lightning events is well corresponded with the radar reflectivity over 30 dBz within the analysis area. As a result, we have the reason to believe that the FTLLS is highly reliable for the detection of lightning events and the result is acceptable for the analysis of the thunderstorm movement.

[Figure]

(a) Estimated horizontal location errors

(b) Estimated vertical location errors

**Figure 9.** The estimated locating errors at an altitude of 10 km ((**a**) horizontal error, (**b**) vertical error).

[Figure]

Figure. The stacking map of reflectivity scan by WSR-98D research radar and spatial distributions of total lightning data on 17 May 2014.

**Minor comments:**
- Line 51-52: Citation?

Authors' Response

Thanks for your comments. We have revised the description of the article we cited.

- Lines 53-55: I don't follow this reasoning

Authors' Response

Thanks for your comments. In the conclusion of the article 'Thunderstorm occurrence and characteristics in Central Europe under different synoptic conditions' (Wapler and James, 2015), it says 'The detailed analysis of convective cell characteristics shows that there is a significant dependence between various cell attributes and the GWLs. E.g., those types associated with broadly westerly flow tend to have high cell speeds and relatively narrow distribution of cell directions. Those Grosswetterlagens which have lower average cell speeds tend to have a higher likelihood of hail.' The Grosswetterlagens in the article means the large-scale weather conditions in Central

Europe. We have revised the description of the article we cited. And we believe the citation is reasonable.

- Line 111: How many thunderstorms are excluded by this condition?

Authors' Response

Thanks for your comments. There is no specific number of thunderstorms excluded by the threshold mentioned in the article. The authors believe that it is unnecessary to study the discarded storm because the less strong thunderstorms are not considered in the article since it is the strong thunderstorms that pose great damage. We mainly focus on eight cases instead of all storms in the PRD region. The discarded thunderstorms could be study separately.

- Lines 124-135: I'm confused where the subscripts 1 and 2 come from. It seems like each storm would receive one Clat and one Clon, so how are two Clat's and two Clon's being calculated to derive the direction and velocity? If this is referencing Clats and Clons from multiple 12-min grids, how were the joined into a single storm?

Authors' Response

Thanks for your comments. As has been explained in the article, we can obtain the coordinate of discharge centroid at each 12-min interval through the 8-adjacent connected-neighborhood labeling method. Two clusters whose discharge centroid is less than 10 km merge as one cluster. Each time the window advances 12 minutes, the cluster is updated and also its centroid. If the distance from previous cluster to the next cluster is less than 50 km, they can be regarded as the same thunderstorm. We obtain the coordinates of all clusters in the area and we choose the thunderstorms whose lifetime is longer than 60 minutes. The tracks are defined by the sequence of centroid positions. The subscripts 1 and 2 represent the discharge center of storm clusters at different times. For example, we calculate the coordinates of discharge centroids at 14:00 and 14:12 respectively. We define the coordinate at 14:12 as the $(C_{lon2}, C_{lat2})$, and the coordinate at 14:00 as the $(C_{lon1}, C_{lat1})$. After getting two coordinates, we can calculate to obtain the direction and velocity. The corresponding explanation has been revised to make it clearer in the manuscript.

- Line 164: What is the significance of comparing each storm's lightning to the rest of the lightning observed by the FTLLS?

Authors' Response to the comments in Line 111 and 164

Thanks for your comments. As the referee has mentioned, there are some storms that have been excluded. However, it is unnecessary to study the discarded storm. It is the total lightning data that we use to characterize the dynamic movement of thunderstorms, this is why we show the number of lightning events excluded by the condition in Figure 3. It shows the result of lightning events excluded by the thresholds. The corresponding explanation has been added to the manuscript.

- Line 128: Normally true North serves as the benchmark. This decision results in some hard-to-interpret graphics later in the paper. For instance, west-to-east moving

storms (as these appear to be from Figure 4), receive directions of ~0 or ~360 degrees, as opposed to 270 degrees that we normally associated with westerly wind.

Authors' Response

The issue pointed out by the reviewer is very important. The authors decide to change the expression of direction to the true North benchmark to better combine the influence of the summer monsoon. The corresponding revision has been made to make it more acceptable in the manuscript.

- Line 141: Are lightning events the same thing as flashes? Or are they strokes? Please clarify in text.

Authors' Response

The authors would like to express our appreciation for the reviewer's suggestions. As we all know, lightning flashes include the IC flashes and CG flashes, and each CG flash consists of one or more leaders followed by one or more return strokes. The FTLLS detects electromagnetic waves associated with lightning discharges and locates VLF/LF (200 Hz–500 kHz) radiation sources. The remote sub-stations acquired triggered waveforms with a duration of 0.5 ms and a resolution of 12-bits. The discharge which meets the threshold will be considered as a lightning event. Therefore, an IC or CG flash consists of several IC or CG events. The type of lightning event is classified by the waveform characteristics and the height information (Cai et.al 2019). The corresponding explanation has been added to the manuscript.

- Line 186, 208: "Severe" storms have a particular meaning (i.e., producing some sort of surface hazard that makes them severe), and surface hazards were not mentioned in the analysis.

Authors' Response

The issue pointed out by the reviewer is very important. In 2014, A total of 12 heavy rainfall events occurred during the flood season, including the period from 15 May ti 23 May. From the night of May 22 to 23, 2014, there were heavy rainstorms in Guangzhou. The center place of precipitation appeared in the PRD region, more specifically, the central and northern parts of Conghua, central and northern Zengcheng, and eastern Huadu. The precipitation in some areas exceeded historical extremes. From May 16 to 17, 2014, severe thunderstorms occurred in most parts of Guangdong accompanied by heavy local rains and strong winds of magnitude 7 to 9.

In view of the reviewer's comment, the following description and figures have been added to the revised version of the manuscript.

*Persistent heavy rainfall occurred from 17 May 2014 to 23 May 2014, especially in the central and eastern part of the PRD region. It was found that 62 automatic weather stations recorded heavy precipitation of more than 100 mm on 17 May, while Huizhou and Shenzhen stations recorded 24-h rainfall of 377. 9 mm and 274. 6 mm (Bingzhi Zheng, 2015). Severe thunderstorms occurred in the PRD region with a record-breaking 24-h rainfall of 477.4 mm starting from 2000 LST (local standard time = UTC + 8 h)*

*22 May during this heavy rainfall week. The hourly precipitation in Conghua station surpassed 60 mm at 1300 LST on 23 May (Xinyu Zhou, 2017; Zhongqing Liang, 2015).*

- Line 287: How do the rivers affect the storms?

Authors' Response

The issue pointed out by the reviewer is very important. The rainy season in the south of China refers to the April–June period, in which precipitation is primarily related to fronts and monsoon flows. It is divided into the pre-monsoon and monsoon period. The latter is affected by the South China Sea Summer Monsoon (SCSSM) which breaks out in May on average. After the SCSSM outbreak (monsoon period), the southwesterly monsoon flow, in particular the southwesterly LLJ, carries abundant water vapor to South China, enhancing precipitation in the region.

A pertinent discussion on this aspect has now been added in the Data and Methodology as follows.

*During the monsoon period (in May on average) in the PRD region, the South China Sea Summer Monsoon (SCSSM) enhances the precipitation owing to the southwesterly monsoon flow, especially the southwesterly low-level jets, carrying abundant water vapor to South China (Chen and Luo, 2018; Bei et al., 2002).*

- Line 288: Figure 4 seems to indicate the storms move from west to east?

Authors' Response

Thanks for your comments. We have revised the clerical error in the Results.

- All – Needs editing and spelling check (e.g., "dimention" and "adjacenct")

Authors' Response

Thanks for your comments. The typing errors have been checked and revised throughout the article.

Betz, H. D., K. Schmidt, W. P. Oettinger, and B. Montag, 2008: Cell-tracking with lightning data from LINET. Adv. Geosci., 17, 55-61, https://doi.org/10.5194/adgeo-17-55-2008.

Cai, L., X. Zou, J. Wang, Q. Li, M. Zhou, and Y. Fan, 2019a: The Foshan Total Lightning Location System in China and Its Initial Operation Results. Atmosphere, 10, https://doi.org/10.3390/atmos10030149.

Cai, L., X. Zou, J. Wang, Q. Li, M. Zhou, Y. Fan, and W. Yu, 2019b: Lightning electric‐field waveforms associated with transmission‐line faults. IET Generation, Transmission & Distribution, 14, 525-531, https://doi.org/10.1049/iet-gtd.2019.0736.

Chen, X., K. Zhao, and M. Xue, 2014: Spatial and temporal characteristics of warm season convection over Pearl River Delta region, China, based on 3 years of operational radar data. Journal of Geophysical Research: Atmospheres, 119, 12,447-412,465.

Chen, X., K. Zhao, M. Xue, B. Zhou, X. Huang, and W. Xu, 2015: Radar‑observed diurnal cycle and propagation of convection over the Pearl River Delta during Mei‑Yu season. Journal of Geophysical Research: Atmospheres, 120, 12557-12575.

Liu, Y., and Coauthors, 2014: Physical and observable characteristics of cloud‑to‑ground lightning over the Pearl River Delta region of South China. Journal of Geophysical Research: Atmospheres, 119, 5986-5999.

Meyer, V. K., H. Höller, and H. D. Betz, 2013: Automated thunderstorm tracking: utilization of three-dimensional lightning and radar data. Atmospheric Chemistry and Physics, 13, 5137-5150, https://doi.org/10.5194/acp-13-5137-2013.

Rigo, T., N. Pineda, and J. Bech, 2010: Analysis of warm season thunderstorms using an object-oriented tracking method based on radar and total lightning data. Natural Hazards and Earth System Sciences, 10, 1881-1893, https://doi.org/10.5194/nhess-10-1881-2010.

Strauss, C., M. B. Rosa, and S. Stephany, 2013: Spatio-temporal clustering and density estimation of lightning data for the tracking of convective events. Atmospheric Research, 134, 87-99, https://doi.org/10.1016/j.atmosres.2013.07.008.

Xu, W., E. J. Zipser, and C. Liu, 2009: Rainfall characteristics and convective properties of mei-yu precipitation systems over South China, Taiwan, and the South China Sea. Part I: TRMM observations. Monthly Weather Review, 137, 4261-4275.

Zhang, C., Y. Huang, B. Mai, and N. Wen, 2014: Temporal and spatial characteristics of lightning in Guangdong region. 2014 International Conference on Lightning Protection (ICLP), IEEE, 1173-1176.

---

## Author Response (AR2)

**Point to point response**

By Jianguo Wang, Si Cheng, Li Cai

First of all, the authors wish to thank all reviewers for the comments which significantly improved the content of the manuscript. The authors have addressed all the comments raised by the reviewer and incorporated them in the revised manuscript wherever required.

**Reviewer Comments:**

**#Reviewer 2**

This paper titled, Characterizing the dynamic movement of thunderstorms using VLF/LF total lightning data over the Pearl River Delta region, describes the use of lightning location data to characterize thunderstorms in various ways. The authors provided detailed discussion of the methodology related to the storm clustering algorithm, with nice visuals to clearly illustrate what was done. The figures are generally well made and professional looking. However, due to my major concerns with the paper that I outline below, I believe there should be major revisions done. If the authors can address my concerns, I do believe this work can be a positive contribution to the literature.

Authors' Response

The authors thank the reviewer for providing all the suggestions and sincerely accept that these have turned out to be indispensable in pushing and improving the standard of the current work.

Major comments:

I am concerned that the authors have not done a thorough literature review for the background of this study. They state that no other studies have used lightning data to study thunderstorm characteristics, which is simply not true. Below I list several after just a few minutes of online research. I think the authors need to take the time to properly review the literature to understand what research has been completed in this domain.

Authors' Response

Thanks for your comments. As the reviewer says, there are indeed many papers that are concerned about the thunderstorm characteristics by using lightning data. However, most of these thunderstorm characteristics researches are about the number of total lightning flashes, the ratio of IC to CG, the ratio of positive CG, the lightning 'jump', the relationship between radar and lightning flashes and so on. Also, the clustering of thunderstorm is mostly based on the radar reflectivity, instead of total lightning data. The main point of our paper is to study the movement of thunderstormsrm by using the total lightning data, including the duration time, valid area, movement velocity, farthest distance and direction. We have made some revisions topinpointt our main purpose. We have also enhanced the literature review and added a more detailed description of the thunderstorm movement characteristics.

My biggest concern with the paper is that much of the discussion relates to statistics (max, min, median, etc.) and makes some generalized claims relating to these quantities. However, the sampling size of this study is extremely small with only 8 storms. Because of this, I think the authors need to refrain from making generalized claims.

Authors' Response

The authors would like to express our appreciation for the reviewer's suggestions. We have revised the generalized claims in the article to make our conclusion more precise.

There seems to be no discussion about whether the results could be influenced or biased by the methodology. The most obvious example relates to the storm velocities (Figure 6). In my opinion, the large variability is more an indication that the 12-min windows are simply too large to properly resolve the fast, but smoothly changing velocity of the storms. There should be more discussion about how there may be biases introduced by the data and/or the methodology.

Authors' Response

Thanks for your comments. The reviewer is right. We have also noticed the change of velocity and already made some discussion in the manuscript. 'In this paper, velocities are calculated by the discharge centroids of the thunderstorm. Owing to the instability of updraft and non-inductive electrification in the convective cloud, the discharge centroid is not always the barycentre of thunderstorm clusters. As the movement metrics are obtained through the lightning events, which represent the electrification in the cloud, the discharge centroid can better reflect the electrification variations in the storm.'

As the reviewer says, this method is very likely to bring about the bias of velocity, leading to a relatively fluctuant variation. The time interval also may cause the fluctuation of velocity. Some severe thunderstorms like supercells last for a short time and move extremely fast. A large time interval may be not able to well monitor the movement of the thunderstorm. We have added the corresponding discussion in the manuscript.

*The fluctuation of velocity is very likely to result from the calculation method of centroid discharge. In addition, the different time intervals may cause the bias of velocity. Some severe thunderstorms like supercells last for a short time and move extremely fast, leading to poor monitoring results.*

Finally, I found that the Discussion and Conclusions section to be significantly lacking discussion of the results. It was often difficult to follow what the authors were trying to point out. The authors need to spend more time expanding the discussion.

Authors' Response

Thanks for your comments. We have made the corresponding revisions in the manuscript based on your comments.

See my line numbered comments below for more details on my major concerns.

Comments by line number:

Line 15: first distribution should be plural

Authors' Response
Thank you very much for your carefulness. The corresponding revision has been made in the manuscript.

Line 51: hail should be singular

Authors' Response
Thank you very much for your carefulness. The corresponding revision has been made in the manuscript.

Lines 58-60: I actually disagree with this statement and feel that the authors did not do a thorough literature review before making this statement. After a few minutes of investigation, I found multiple papers that specifically look at the lightning characteristics over the course of individual thunderstorms. Here are a few that I was able to find within a few minutes of online searching.

Soula, S., & Chauzy, S. (2001). Some aspects of the correlation between lightning and rain activities in thunderstorms. Atmospheric research, 56(1-4), 355-373.

Williams, E., Boldi, B., Matlin, A., Weber, M., Hodanish, S., Sharp, D., ... & Buechler, D. (1999). The behavior of total lightning activity in severe Florida thunderstorms. Atmospheric Research, 51(3-4), 245-265.

Wang, C., Zheng, D., Zhang, Y., & Liu, L. (2017). Relationship between lightning activity and vertical airflow characteristics in thunderstorms. Atmospheric Research, 191, 12-19.

Authors' Response
Thanks for your comments. Indeed, there are many articles about lightning characteristics over the course of individual thunderstorms. However, these characteristics are mostly about the number of total lightning, the lightning 'jump' and the relationship between radar and lightning flashes. Seldom papers concentrate on the kinematic characteristics of every single thunderstorm, such as the direction and speed of movement or cell sizes and severity.
The article you mentioned above 'Wang, C., Zheng, D., Zhang, Y., & Liu, L. (2017). The relationship between lightning activity and vertical airflow characteristics in thunderstorms.' is related to the thunderstorm identification which is a meaningful reference for our paper. But, it is not concerned about the movement of thunderstorms and it is an analysis of the overall 22 storms from 2010 to 2012, instead of the individual thunderstorm.
We put forward five kinematic parameters (duration time, valid area, movement velocity, farthest distance and direction) to quantify the movement of clusters in various periods of a day. This is exactly the key point of this paper. We have revised to explain this.

*However, up to now, there have been few formal studies that individually analyze such fundamental ==kinematic characteristics== of every single thunderstorm.*

Line 80: The use of heights for classifying IC versus CG flashes is problematic and can lead to many misclassified CGs as ICs. Combining this with waveform classification will improve this, however, this is very little information provided regarding the details or the efficacy of the algorithm. Can the authors provide some literature review as to how accurate this method is as well as the expected false classification rate?

Authors' Response

Thanks for your comments. We have stated in the manuscript that 'The validation of the system has been guaranteed through the comparison of rocket-triggered lightning experiments and the application of transmission lines (Cai et al., 2019; Wang et al., 2019).'

Specifically, from the initial operation result, it has shown good three-dimensional location accuracy and detection efficiency. A total of 27 lightning strokes, with 168 field waveforms included, were recorded by FTLLS to explain the transmission-line fault in the Foshan area (Wang et al., 2019). In the rocket-triggered lightning experiment in Guangzhou, 38 return strokes in six triggered flashes were observed by nine sites of FTLLS to compare the differences between rocket-triggered lightning and subsequent return strokes in natural flashes(Cai et al., 2019).

In addition, the detection efficiencies and peak-current estimation of the FTLLS and the LLS of Guangdong power grid in Guangdong Province were examined and were compared based on the directly measured current data at the triggering lightning site. It is shown that the detection efficiencies of the FTLLS for flashes and for return-strokes were 87.5% (7/8) and 93.0% (40/43), respectively. The peak-current estimation error reported by the FTLLS was 8% on average (Li et al., 2021).

We have added the following reference to explain the accuracy of FTLLS in the manuscript.

*==Li et al. (2021) examined the detection efficiencies using the directly measured current data at the triggering lightning site. The result shows that the detection efficiencies of FTLLS for flashes and for return strokes were 87.5% and 93.0%, respectively.==*

Line 82: This type of network does not inherently detect lightning flashes. Because it is an VLF/LF network, it is detecting charge motion occurring from current pulses. Therefore, there must be some pulse clustering occurring to clustering these pulses into flashes. This clustering methodology is important to this analysis since it can impact the number of flashes and flash rates. Therefore, the authors need to describe the flash clustering algorithm used in the paper.

Authors' Response

Thanks for your comments. The reviewer is right. We cannot directly get the number of flashes from the FTLLS. The current pulse detected by the FTLLS in this paper is defined as a lightning event. However, the main aim of this paper is to study the movement of the thunderstorm. We do not need to cluster the lightning events into flashes and still can get the centroid and valid area of the thunderstorm. We have added the definition of the lightning event in 'Foshan total lightning location system' part.

As for Figure 4, it not only shows the variation of the number of lightning events in each thunderstorm, but also reflects the difference between thunderstorms. As the

FTLLS is the new system that can first detect the total lightning events over the Pearl River Delta region. We believe that more studies about the flash clustering algorithm and the comparison with other LLS can be done in the future.

Line 89: Cai et al., **2019**
Authors' Response
Thanks for your comments. We have revised the mistake.

Line 109: wording here is unclear, please rephrase.
Authors' Response
Thanks for your comments. We have revised the sentence as follows.

*The analysis of total lightning is in progress with a 12-minute time interval, which was twice of the Doppler Radar scans.*

Lines 137-142: There seems to be no way for a thunderstorm to split and may affect the results since this does happen in real thunderstorms. Have the authors considered this? I think this is worth discussing in the paper.
Authors' Response
Thanks for your comments. We have added the split situation of the thunderstorm in the '2.2 Data and Meterology' part. The process of the thunderstorm is shown as follows.

*As the coordinates of discharge centroids within a time interval are obtained, two clusters whose discharge centroid is less than 10 km merge as one cluster. For the split of the thunderstorm, if there is more than one cluster within the analysis region, the cluster with the largest area is set as the main body of the thunderstorm. The cluster whose discharge centroid is more than 10 km away from the main cluster's discharge centroid will be seen as the split part of the thunderstorm and be discarded.*

Lines 170-171: Check grammar.
Authors' Response
Thanks for your comments. We have revised the sentence as follows.

*The midday and afternoon thunderstorms keep relatively strong and stable, with more than twelve thousand lightning events per hour, while the morning thunderstorms are much more gentle and weaker, with around ten thousand lightning events per hour.*

Line 190: This phrasing makes it seem like the thunderstorm initiation and dissipation is decided by radar data. Is that true? If so, the previous methodology made it sound like the lightning was the only deciding factor for clustering flashes into thunderstorms. If not, please rephrase.
Authors' Response
Thanks for your comments. We have added the definition of the starting time and ending time of the thunderstorm in 'Data and Methodology' part. The thunderstorm started with the appearance of the valid area ($>25$ km$^2$), while the ending time of the thunderstorm is when the cluster is less than 25 km$^2$ and can not be depicted by the algorithm any longer. So, the analysis period of the thunderstorm we choose is not decided by radar data. The reason why we mention the radar data is to reconfirm that the lightning data is reliable. The corresponding revision is as follows.

*The thunderstorm started with the appearance of the valid area (>25 km², while the ending time of the thunderstorm is when the cluster is less than 25 km² and can not be depicted by the algorithm any longer.*

Line 191: You should state explicitly that the IC and CGs are for the storm only, not the entire dataset.

Authors' Response

Thanks for your comments. We have added the relative description in the manuscript as follows.

*The blue line and red lines represent the IC events and CG events ==produced by the chosen thunderstorm==, respectively.*

Lines 192-194: I do not agree with this statement. It is true that the CG rate usually peaks near the middle to end, but so does the IC rate. Maybe the authors meant to say that the IC:CG ratio is highest at the beginning of the storm? I am not sure that is true either, but to show that you could add a curve for the IC:CG ratio on the plots as well.

Authors' Response

Thanks for your comments. Indeed, we cannot compare the occurrence between the IC peak and the CG peak. There is also no obvious evidence that the IC:CG ratio is highest at the beginning of the storm. So, we revise the conclusion from Figure 4 as follows.

==*The thunderstorm occurred at 18:00 on 17 May produced the largest number of total lightning events per 12 minutes, with the number being more than 8000 times. Another night storm occurred at 19:12 on 19 May is much weaker, whose scale is slightly smaller than the morning storm occurred at 8:36 on 20 May, with a smaller peak of total lightning per hour and a smaller number of total lightning events.*==

Line 196: "visions to the horizon" does not make sense. Maybe "The footprint, trajectory, and flash density of thunderstorms" is more accurate?

Authors' Response

Thanks for your comments. The reviewer is right. We have revised the corresponding description as follows.

*==The footprint, trajectory, and flash density of thunderstorms== are displayed in Figure 5.*

Line 225-229: I find the discussion of the velocity results to be somewhat biased in that there is no discussion that they could be a result of the method or sampling of the methodology. When observing the motion of a storm in real-time, there is no apparent drastic changes in motion of typical storms. They generally move quite smoothly. I believe the result that the velocity seems to vary so much is more related to the methodology. It could indicate that the 12-min windowing is sampling the motion of the storm too slowly, resulting in these results not being able to resolve the proper smooth changes of the storm.

Authors' Response

Thanks for your comments. As we have mentioned above. We have also noticed the change of velocity and made the corresponding discussion in the manuscript.

*Owing to the instability of updraft and non-inductive electrification in the convective cloud, the discharge centroid is not always the barycentre of thunderstorm clusters. As the movement metrics are obtained through the lightning events, which represent the electrification in the cloud, the discharge centroid can better reflect the electrification variations in the storm. The storm with the highest speed occurred on the morning of 20 May, with a value of 204.8 km/h. The lowest maximum speed was 115.3 km/h occurred on the evening of 19 May. The velocity does not show the same tendency as the variation of VA during the lifetime of thunderstorms. It oscillates severely compared with the valid area which shows a steady increase or decrease during the lifetime of thunderstorms. The fluctuation of velocity is very likely to result from the calculation method of centroid discharge. In addition, the different time intervals may cause the bias of velocity. Some severe thunderstorms like supercells last for a short time and move extremely fast, leading to poor monitoring results. A relatively large velocity variation is also seen in the Mediterranean storm (Betz et al., 2008), but with a general upward trend in some cells during the whole movement. Meyer et al. (2013) proposed that long-lived storms are most likely fast propagation as the storms with velocities around 80 km/h spent 150 min to 240 min to cross the domain, however, this was under-represented because of the insufficient statistics. The eight cases in this study also do not show this trend.*

Lines 268-272: These references all relate to radar data and seem inappropriate to reference the characteristics used for lightning data. As I mentioned above, there are previous papers that have used lightning data to estimate thunderstorm characteristics.

Authors' Response

Thanks for your comments. The purpose of this paper is to analyze the movement characteristics of thunderstorms, instead of the characteristics of lightning itself. The thunderstorm evolution and movement can be obtained both by the lightning data and radar data. Meyer et al. (2013) provide a hybrid method that combines the radar data and lightning data to assess, track, and monitor a more comprehensive picture of thunderstorms. We believe that it is necessary to compare and discuss the kinematics features of thunderstorms with various data sources. And it also enriches the method to analyze thunderstorm movement.

In Line 268-272, we have discussed the different thresholds used in the literature. Some papers used radar data or both radar and lightning data to identify the thunderstorm with different thresholds including the radar reflectivity, pixel area and duration. The pixel area and duration thresholds are also used in our paper. It is worthwhile to discuss the differences.

Line 286: Please re-state what the mean was for this study so that the reader does not need recall what it was or to go back and find it.

Authors' Response

Thanks for your comments. We have revised the corresponding description as follows.

*The lifetime was between 54 min to approximately 8h, with the average thunderstorm duration of the whole evolution process being about 3.5 h, which is slightly longer than this study.*

Line 287: 6-minute time interval? I thought the time intervals for this paper were 12-minutes.

Authors' Response

Thank you very much for your carefulness. The sentence 'The average area was 509 km$^2$ in a 6-minute time interval, with the biggest cluster area in the mature stage.' in line 287 is the description of the literature. It is the finding of Rigo et al. (2010) who analyzed the thunderstorm at the 6-minute time interval.

Line 289: This is not a discussion, this is simply stating what others have found. Please include discussion on how this compares to the current study. It sounds like the storms in this study had an average area equal to the maximum area of a supercell (which is defined as an extremely strong thunderstorm). That seems very unlikely. This raises several concerns related to the methodology of the paper. The authors need to provide much more detailed discussion.

Authors' Response

Thanks for your comments. The maximum cell area of the June supercell reported by Meyer et al. (2013) was found to be nearly 500 km$^2$ in the 3-min interval. We have mistakenly treated it as 15 minutes, which is actually very unlikely. The average area of 66 Catalonia warm-season thunderstorms was 509 km$^2$ in a 6-minute time interval (Rigo et al., 2010). In our study, the average area of eight thunderstorms is 336 km$^2$ per 12 minutes. The differences derive from the geographic position, the severity of thunderstorms and the clustering methodology. We have added the discussion in the manuscript.

*Rigo et al. (2010) reported the duration and the average area of 66 Catalonia warm-season thunderstorms. The lifetime was between 54 min to approximately 8h, with the average duration of the whole thunderstorm evolution process being about 3.5 h, which is slightly longer than this study. The average area of 66 thunderstorms was 509 km$^2$ in a 6-minute time interval, with the biggest cluster area in the mature stage. A June supercell propagated north of Munich in the eastern direction was reported by Meyer et al. (2013) to illustrate the area, velocity and farthest distance of storms, showing that the maximum cell area was nearly 500 km$^2$ in the 3-min interval. The average area of eight thunderstorms in this paper is 336 km$^2$ per 12 min. The differences derive from the geographic position, the severity of thunderstorms, the clustering methodology and so forth.*

Line 294-295: How is this conclusion found? Once again, there is not enough discussion provided for the reader to follow this statement.

Authors' Response

Thanks for your comments. We have added the specific date of Meyer's case. The conclusion was from Figure 9 of Meyer et al. (2013). It shows the lifetime of selected cell parameters for the thunderstorm case study from 25 June 2008. We can see that the area variation of Meyer's case occurred on 25 June 2008 appears to be more fluctuant with a sharp decrease in the developing stage and many peaks during the whole evolution process. The corresponding revision had been shown as follows.

[Figure]

**Fig. 9.** Lifetime diagrams of selected cell parameters for the thunderstorm case study from 25 June 2008. **(a)** Radar- and lightning-cell areas (top) and iso-reflectivity heights a.g.l. (green) and overshooting top height relative to the mean anvil top height estimated from POLDIRAD radar data (grey bars) (bottom, shared axis). **(b)** 3 min TL stroke rates per cell (top) and per respective lightning-cell area (bottom). **(c)** IC fraction to TL stroke rate (top) and precipitation "intensity" per radar-cell area as described in text (bottom).

*The area variation of Meyer's case occurred* *on 25 June 2008* *appears to be more fluctuant, with a sharp decrease in the developing stage and many peaks during the whole evolution process.*

Line 304: "Owning" should be "Owing"

Authors' Response

Thank you very much for your carefulness. The corresponding revision has been made in the manuscript.

Line 318: Are the authors sure that the direction angle are in the same reference frame as this study? Storms in the USA also generally move from West to East.

Authors' Response

Thanks for your comments. We have deleted contents about the direction of thunderstorm in the USA and added possible causes of the thunderstorm movements over the PRD region. We believe that the influence factors are more relevant and worthwhile to be disccused in this paper. The discussion is shown as follows.

*The orientation of thunderstorms can be affected by the topographic relief (Miller and Mote, 2017). Lin et al. (2011) found that the warm season afternoon thunderstorm over Taiwan Island frequently occurred in a narrow strip, parallel to the orientation of the mountains, along the lower slopes of the mountains. The urban heat island effects and northern mountains in Guangzhou city may influence the movement of thunderstorms over the PRD region (Yin et al., 2020).*

Figure 1: The colormap label is confusing. Seems to me the top two row colors are exactly the same or extremely similar. Please consider redoing it.

Authors' Response

Thanks for your comments. We have changed the top two row colors in Figure 1.

[Figure]

Figure 2: 16:00 is missing. Is that intentional? That is confusing and should be stated in the caption for readers to understand.

Authors' Response

Thanks for your comments. Considering that two thunderstorms have been chosen for study on 17 May (one is from 11:36 to 14:36, and another is from 17:48 to 20:36), we put the stacking map from 12:00 to 15:00 and 18:00 to 21:00 to display the location and variation of the thunderstorm. We have added the corresponding explanation in the manuscript as follows.

*Figure 2 showed general radar characteristics and lightning distributions of thunderstorms from 11:36 to 14:36 and 17:48 to 20:36 on 17 May.*

Figure 4: "during the thunderstorm process", it is unclear to me what you mean by this. Do you mean the lightning associated within each chosen thunderstorm? please rephrase for clarity.

Authors' Response

Thanks for your comments. We have revised the description in the manuscript as follows.

*The light blue shaded areas represent total lightning events ==produced by the chosen thunderstorm.==*

Figure 5. Similar to Figure 4, you should add the Storm ID so that readers can easily compare. Also, the caption is wrong. The horizontal is actually longitude, while the vertical is latitude.

Authors' Response
Thank you very much for your carefulness. The corresponding revision has been made in the manuscript as follows. The Storm ID of Figure 5 is longer than that of Figure 4 because it includes the specific time. So, we put the Storm ID on the top of the figure.

*The horizontal axis corresponds to the longitude with the vertical axis standing for the latitude.*

[Figure]

Betz, H. D., Schmidt, K., Oettinger, W. P., and Montag, B.: Cell-tracking with lightning data from LINET, Advances in Geosciences, 17, 55-61, 2008.

Cai, L., Zou, X., Wang, J., Li, Q., Zhou, M., Fan, Y., and Yu, W.: Lightning electric-field waveforms associated with transmission-line faults, IET Generation, Transmission & Distribution, 14, 525-531, 10.1049/iet-gtd.2019.0736, 2019.

Li, Q., Wang, J., Cai, L., Zhou, M., and Fan, Y.: On the return-stroke current estimation of Foshan Total Lightning Location System (FTLLS), Atmospheric Research, 248, 105194, https://doi.org/10.1016/j.atmosres.2020.105194, 2021.

Lin, P.-F., Chang, P.-L., Jou, B. J.-D., Wilson, J. W., and Roberts, R. D.: Warm Season Afternoon Thunderstorm Characteristics under Weak Synoptic-Scale Forcing over Taiwan Island, Weather and Forecasting, 26, 44-60, 10.1175/2010waf2222386.1, 2011.

Meyer, V. K., Höller, H., and Betz, H. D.: Automated thunderstorm tracking: utilization of three-dimensional lightning and radar data, Atmospheric Chemistry and Physics, 13, 5137-5150, 10.5194/acp-13-5137-2013, 2013.

Miller, P. W. and Mote, T. L.: A Climatology of Weakly Forced and Pulse Thunderstorms in the Southeast United States, Journal of Applied Meteorology and Climatology, 56, 3017-3033, 10.1175/jamc-d-17-0005.1, 2017.

Rigo, T., Pineda, N., and Bech, J.: Analysis of warm season thunderstorms using an object-oriented tracking method based on radar and total lightning data, Natural Hazards and Earth System Sciences, 10, 1881-1893, 10.5194/nhess-10-1881-2010, 2010.

Wang, J., Li, Q., Cai, L., Zhou, M., Fan, Y., Xiao, J., and Sunjerga, A.: Multiple-Station Measurements of a Return-Stroke Electric Field From Rocket-Triggered Lightning at Distances of 68–126 km, IEEE Transactions on Electromagnetic Compatibility, 61, 440-448, 10.1109/temc.2018.2821193, 2019.

Yin, J., Zhang, D.-L., Luo, Y., and Ma, R.: On the Extreme Rainfall Event of 7 May 2017 over the Coastal City of Guangzhou. Part I: Impacts of Urbanization and Orography, Monthly Weather Review, 148, 955-979, 10.1175/mwr-d-19-0212.1, 2020.

---

## Author Response (AR3)

**Point-to-point response**

By Jianguo Wang, Si Cheng, Li Cai

First of all, the authors wish to thank all reviewers for the comments which significantly improved the content of the manuscript. The authors have addressed all the comments raised by the reviewer and incorporated them in the revised manuscript wherever required.

**Reviewer Comments:**

**#Reviewer 2**

This paper titled "Characterizing the dynamic movement of thunderstorms using VLF/LF total lightning data over the Pearl River Delta region" discusses the use of lightning data to track and analyze the movement of thunderstorms in Southern China. The authors did a nice job responding to all of my comments. The paper looks much better and is close to ready for publication. However, their comments did reveal one important issue that needs to be addressed before it can be published. There I suggest minor revisions.

Authors' Response

The authors thank the reviewer for providing all the suggestions and sincerely accept that these have turned out to be indispensable in pushing and improving the standard of the current work.

In the response to my comment regarding flash clustering information, the authors state that they do not do any flash clustering and use the individual pulses detected by the FTLLS in their thunderstorm data. However, on many occasions they are using "flash" (e.g., lines 111, 135, 206, 284). If there is no flash clustering occurring, then this is not an accurate statement. They authors should use event, not flash. Furthermore, this means that this study cannot compare to other studies that use flash rate/total since there are many pulses per flash. The authors need to read through the paper and eliminate any use of flash when referencing this dataset.

Authors' Response

The authors would like to express our appreciation for the reviewer's suggestions. We have read through the paper and revised the corresponding description of the lightning event. We have also deleted the flash-related reference in the manuscript.

Comments by line #

Line 90: My previous comment was not related to the detection efficiency of the system, but the classification accuracy. Since there are results that use IC vs CG, I would like to see some results related to the classification accuracy.

Authors' Response

Thanks for your comments. The classification accuracy of a total lightning location system is usually obtained by comparing it with other total lightning location systems. However, as we have stated, the FTLLS in this study is the first total lightning location system over the PRD region. So, there is no other LLS to compare with. We believe

that more total lightning location systems will be established in China and related research will be done in the future. Meanwhile, we have already shown that the detection of CG is accurate by comparing it with the triggering lightning result and transmission line fault (Line 89-91).
In addition, the classification accuracy of IC and CG has little influence on the result of the thunderstorm movement, because the tracking of thunderstorms was done by the total lightning events.

Line 113: This wording is still not clear. " is in progress " sounds like it is not complete and will be done in the future.
Authors' Response
Thanks for your comments. We have revised the sentence to make it clearer.

*The time interval of 12 min is twice of the Doppler Radar scans, with which the routes of thunderstorms can be tracked precisely without losing kinematic features.*

Line 296: You still do not state what the average is for the current study, which would make it much easier for the reader to compare rather than having to look back in the results.
Authors' Response
Thanks for your comments. We have added the average duration of thunderstorms in this study.

*The lifetime was between 54 minutes to approximately 8 hours, with the average duration of the whole thunderstorm evolution process being about 3.5 h, which is slightly longer than this study (2.93 hours)*